# Mitigating Graph Covariate Shift via Score-based Out-of-distribution Augmentation

## Abstract

Distribution shifts between training and testing datasets significantly impair the model performance on graph learning. A commonly-taken causal view in graph invariant learning suggests that stable predictive features of graphs are causally associated with labels, whereas varying environmental features lead to distribution shifts. In particular, covariate shifts caused by unseen environments in test graphs underscore the critical need for out-of-distribution (OOD) generalization. Existing graph augmentation methods designed to address the covariate shift often disentangle the stable and environmental features in the input space, and selectively perturb or mixup the environmental features. However, such perturbation-based methods heavily rely on an accurate separation of stable and environmental features, and their exploration ability is confined to existing environmental features in the training distribution. To overcome these limitations, we introduce a novel approach using score-based graph generation strategies that synthesize unseen environmental features while preserving the validity and stable features of overall graph patterns. Our comprehensive empirical evaluations demonstrate the enhanced effectiveness of our method in improving graph OOD generalization.

## 1 Introduction

Deep learning algorithms have become predominant in the analysis of graph-structured data. However, a common limitation of existing methods is the assumption that both training and testing graphs are independently and identically distributed (i.i.d.). This assumption often falls short in real-world scenarios, where shifts in data distribution frequently occur, leading to significant degradation in model performance. As a result, there has been considerable progress in improving graph out-of-distribution (OOD) generalization, evidenced by advancements in invariant graph learning (Chen et al., 2022; Wu et al., 2022; Huang et al., 2024) and graph data augmentation (Rong et al., 2019; Wang et al., 2021; Han et al., 2022; Yao et al., 2022; Sui et al., 2024; Li et al., 2024).

Recent studies (Gui et al., 2022; Sui et al., 2024) have identified two primary types of distribution shifts. *Correlation shifts* occur when the statistical relationships between environments and labels differ between the training and testing datasets, assuming that the test environments are represented within the training dataset. *Covariate shifts*, on the other hand, arise when the test environments are not present in the training dataset. A prevalent causal perspective in graph invariant learning (Chen et al., 2022; Wu et al., 2022) suggests that stable features of graphs, which causally determine labels, remain invariant across different environments, whereas varying environmental features contribute to distribution shifts. Consequently, previous studies (Miao et al., 2022; Chen et al., 2022; Wu et al., 2022) have primarily focused on correlation shifts by isolating invariant stable graph patterns from the environmental features. In this work, we focus on the relatively neglected but challenging problem of covariate shift in graph learning.

While different graph data augmentation techniques have been proposed to generate new data across various domains (Rong et al., 2019; Han et al., 2022; Sui et al., 2024), these methods mainly modify existing data within the training set by mixing or dropping edges, which either have limited environmental exploration ability or could result in invalid data samples (e.g., molecular graphs violating chemical rules). Moreover, indiscriminate augmentations (Rong et al., 2019) can distort stable patterns, resulting in uncontrollable augmented distributions. While controlled augmentations (Sui et al., 2024) have shown promising outcomes, they heavily depend on the accurate separation of

stable and environmental patterns, which remains a nontrivial challenge and could be inherently infeasible. These observations prompt an essential inquiry: "*Is it possible to explore the training distribution under control, such that the exploration extends beyond the confines of the training environments while still preserving essential stable patterns?*"

In response, this work introduces an innovative score-based graph augmentation strategy that mitigates graph covariate shift by improving the exploration of training distribution while preserving stable predictive patterns on the generated graphs. In high level, we formulate the problem of OOD data augmentation as graph generation simultaneously conditioned on graph labels and exploration variables, based on the graph generation hypothesis widely used in prior studies (Wu et al., 2022; Yang et al., 2022; Gui et al., 2022; Chen et al., 2022). Specifically, we employ a score-based diffusion probabilistic model, commonly known as a diffusion model (Song et al., 2020), to effectively capture the data distribution of unlabeled graphs. During the generation phase, we introduce a novel guidance scheme that generates augmented graphs, concurrently retaining predictive stable patterns and incorporating explored environments. Our proposed *Out-of-Distribution Diffusion Augmentation (OODA)* framework utilizes graph labels to guide the sampling process toward graphs that are highly likely to contain stable patterns. The exploration parameter facilitates exploration beyond the training graph space by flexibly adjusting the discrepancy from the training distribution. The robustness of the score-based diffusion model ensures the validity of the generated graphs, preventing the formation of invalid structures (e.g., non-viable molecules) that could impair downstream classification performance. Furthermore, our guidance scheme eliminates the need to explicitly split graphs into stable and environmental subgraphs. We experimentally validate our method on both synthetic and real-world graph classification tasks under diverse covariate shift settings. Our results demonstrate that OODA outperforms state-of-the-art baselines, including invariant graph learning and graph data augmentation, highlighting its effectiveness in exploring environments under control while preserving stable patterns.

Our main contributions are summarized as follows:

- We propose a novel graph generation-based environment augmentation approach to address covariate distribution shifts in graph learning. Our method enables controlled exploration of environmental patterns while preserving stable patterns, without the need to explicitly separate them.

- Our approach can simultaneously generate out-of-distribution (OOD) graph structures, node features, and edge features, making it uniquely capable of handling covariate shifts in both feature and structural distributions, as well as when these shifts occur simultaneously.

- Extensive empirical evaluations demonstrate that our framework outperforms state-of-the-art graph OOD generalization methods across diverse tasks, including synthetic, semi-artificial, real-world molecular, and natural language sentiment analysis datasets.

## 2 RELATED WORK

Graph-structured data are inherently complex, characterized by the intricate challenges of irregularity and nuanced structural information. This complexity gives rise to graph out-of-distribution (OOD) problems that not only necessitate addressing shifts in feature distributions but also demand attention to variations of structural distributions. In this context, we summarize two principal categories of algorithms for graph OOD robustness: (i) *invariant graph learning* strategies, which aim to ensure model stability across varying distributions; and (ii) *graph data augmentation* techniques, designed to enhance model generalizability by simulating diverse distribution scenarios.

**Invariant Graph Learning.** The concept of invariant graph learning draws inspiration from seminal works such as those by (Arjovsky et al., 2019; Rosenfeld et al., 2020; Ahuja et al., 2021). This approach aims at identifying stable graph structures (e.g., subgraphs) or representations (predictors) that remain consistent across different environments, thereby enhancing out-of-distribution (OOD) generalization. This is achieved by capturing salient graph features and minimizing empirical risks across varying conditions. In scenarios where establishing causality is complex or where strong assumptions may not hold, the task can be approximated by identifying features that demonstrate invariance under distributional shifts, thereby facilitating OOD generalization (Li et al., 2022). Effective OOD generalization is achieved by basing predictions solely on invariant information (Li et al., 2022). For example, DIR (Wu et al., 2022) distinguishes between invariant and environment-specific

subgraphs by creating varied interventional distributions on the training distribution. CIGA (Chen et al., 2022) further explores this domain by employing synthetic environments and the graph generation process to identify stable features under various distribution shifts. However, this line of research assumes *access to test environments during training*, which is an unrealistic assumption given the impracticality of covering all possible test scenarios. Training in limited environments reduces spurious correlations but fails to generalize to new, unseen environments. DISGEN (Huang et al., 2024) gains promising results in disentangling the size factors from graph representations by minimizing the shared information between size- and task-related information, however, the technique is constrained to handle size generalization. In this work, we propose a framework capable of generalizing to unseen environments characterized by differences not only in graph size but also in graph structure, node features, and edge features.

**Graph Data Augmentation.** Beyond invariant graph learning, graph data augmentation aims to diversify the training distribution, thereby enhancing the out-of-distribution (OOD) generalization of models. DropEdge (Rong et al., 2019) introduces randomness by selectively removing edges, thus varying the training data's structure. M-Mixup (Wang et al., 2021) enriches the dataset by interpolating diverse and irregular graphs within semantic space. $\mathcal{G}$-Mixup (Han et al., 2022) extends this concept to graph classification, interpolating across different graph generators (graphons) to produce augmented graphs. Adversarial augmentation techniques, such as FLAG (Kong et al., 2022), apply gradient-based perturbations to node features, and AIA (Sui et al., 2024) generates adversarial masks on graphs, both aimed at probing environmental discrepancies. Despite these advancements, overcoming the limitations in environmental exploration caused by modifying graphs within the original training set remains a challenge, indicating ongoing opportunities for innovation in graph data augmentation strategies. Recently, environment-aware augmentation frameworks (Li et al., 2024) have utilized environment information to linearly explore training graph structures and node features, however, they depend heavily on high-quality and sufficiently diverse environment information. In practice, annotating environment labels or capturing diverse environment information is costly and often infeasible. In this work, we introduce a generation-based augmentation method that eliminates the need for accessing environment information.

## 3 PROBLEM FORMULATION

**Notations.** We represent a graph with $n$ nodes as $G = (\boldsymbol{A}, \boldsymbol{X}, \boldsymbol{E})$, where $\boldsymbol{A} \in \mathbb{R}^{n \times n}$ is the adjacency matrix, $\boldsymbol{X} \in \mathbb{R}^{n \times a}$ denotes $a$-dimensional node features and $\boldsymbol{E} \in \mathbb{R}^{n \times n \times b}$ encodes $b$-dimensional edge features. Without the loss of generality, we focus on the graph classification task where each graph $G$ is associated with a label $Y \in \mathcal{Y}$, determined by a predefined labelling rule $\mathcal{G} \to \mathcal{Y}$. Following invariant learning (Ahuja et al., 2021; Chen et al., 2022), we denote the graph dataset as $\mathcal{D} = \{(G_i^e, Y_i^e)\}_{e \in \mathcal{E}_{\text{all}}}$, where $(G_i^e, Y_i^e) \sim P_e(G, Y)$ is an i.i.d. draw in the environment $e$ sampled from all possible environments $\mathcal{E}_{\text{all}}$. The complete dataset can be partitioned into a training set $\mathcal{D}_{\text{tr}} = \{(G_i^e, Y_i^e)\}_{e \in \mathcal{E}_{\text{tr}}}$ and a test set $\mathcal{D}_{\text{te}} = \{(G_i^e, Y_i^e)\}_{e \in \mathcal{E}_{\text{te}}}$, where $\mathcal{E}_{\text{tr}}$ and $\mathcal{E}_{\text{te}}$ index the training and testing environments, respectively. In practice, environment information may not be explicitly given, and we further denote the training distribution as $P_{\text{tr}}(G, Y)$ and the testing distribution as $P_{\text{te}}(G, Y)$.

**Graph Classification under Covariate Shift.** With only observing the training set $\mathcal{D}_{\text{tr}}$ sampled from the training distribution $P_{\text{tr}}$ in training environments $\mathcal{E}_{\text{tr}}$, our generalization objective under graph covariate shift is to train an optimal graph classifier $f : \mathcal{G} \to \mathcal{Y}$ that performs well across any possible environments $\mathcal{E}_{\text{all}} \supseteq \mathcal{E}_{\text{tr}}$. We formulate this goal as the following minimization problem:

$$\min_f \mathbb{E}_{e \in \mathcal{E}_{\text{all}}} \mathbb{E}_{(G^e, Y^e) \sim P_e(G, Y)}[\ell(f(G^e), Y^e)], \tag{1}$$

where $\ell(\cdot, \cdot)$ denotes the loss function for graph classification and the expectation is with respect to graphs under all possible environments. However in practice, the training environments $\mathcal{E}_{\text{tr}}$ may not cover all environments, causing degraded classification performance when applying the learned classifier in unseen test environments. This covariate shift calls for an effective manner to sufficiently explore unseen data distribution or environments during model training. We summarize and discuss the various types of graph covariate shifts in detail in Appendix A.2.

**Issues in Graph Augmentation via Environmental Exploration.** To augment the training distribution for mitigating graph covariate shift, existing solutions often approach Eq. (1) by separating

and augmenting the environments: $\min_f \mathbb{E}_{e \in \{\mathcal{E}_{\text{tr}} \cup \mathcal{E}_{\text{aug}}\}} \mathbb{E}_{(G^e, Y^e) \sim P_e(G,Y)} [\ell(f(G^e), Y^e)]$, where the augmented environments $\mathcal{E}_{\text{aug}}$ are obtained based on either interpolating explicitly given environmental labels (Li et al., 2024) or perturbing implicitly separated environmental components (Sui et al., 2024; Wu et al., 2022; Miao et al., 2022; Chen et al., 2022). Obtaining accurate environmental labels and components itself could be high-cost and nontrivial tasks, and separating environmental components could be inherently unfeasible (Chen et al., 2024), which limit the practicability of such strategy. In addition, the subgraph perturbations based augmentation is mainly operated by edge dropping (Sui et al., 2024; Rong et al., 2019) and mixup (Han et al., 2022), which is confined to existing subgraphs in training data and could cause invalid samples (e.g., generating molecules that are chemically invalid).

**Distribution Augmentation with OOD Control.** This work overcomes these limitations by formulating the augmentation problem as a generation-based graph OOD augmentation strategy, which directly models and augments the training distribution, without explicitly requiring the knowledge or separation of environmental information. Specifically, we target on synthesizing an augmented training distribution $\tilde{P}_{\text{tr}}(G, Y)$, which is combined with the original training distribution to obtain the classifier, stated as:

$$\min_f \mathbb{E}_{(G,Y) \sim \{P_{\text{tr}}(G,Y) \cup \tilde{P}_{\text{tr}}(G,Y)\}} [\ell(f(G), Y)]. \tag{2}$$

The augmented distribution $\tilde{P}_{\text{tr}}(G, Y)$ can be implemented in multiple ways, but it needs to satisfy two principles: (1) $\tilde{P}_{\text{tr}}(G, Y)$ should deviate from $P_{\text{tr}}(G, Y)$ in a controlled manner, and (2) the explored graphs in $\tilde{P}_{\text{tr}}(G, Y)$ should preserve the stable patterns of graphs in $P_{\text{tr}}(G, Y)$. However, current graph generation models (Jo et al., 2022; Martinkus et al., 2022; Vignac et al., 2022) cannot directly generate graphs that meet these two criteria. To address this, we propose a novel score-based generative model in Section 4 that captures the augmented distribution $\tilde{P}_{\text{tr}}(G, Y)$ while adhering to both principles.

## 4 SCORE-BASED OUT-OF-DISTRIBUTION GRAPH AUGMENTATION

In this section, we present the novel score-based graph augmentation framework, OODA, designed to generate augmented graphs that retain predictable stable features while also exploring new environments. We begin by discussing the score-based generative model for unlabeled graphs and then extend the model to handle out-of-distribution scenarios with controlled adaptation. Thereafter, we illustrate the working principles and implementation details of OODA.

**Motivation** From the perspective of graph generation, the goal of exploring the training distribution $P_{\text{tr}}(G, Y)$ is to generate OOD graph samples from the conditional distribution $P_{\text{tr}}(G, Y \mid \mathbf{y}_{\text{ood}})$ where $\mathbf{y}_{\text{ood}}$ represents the OOD exploration condition. We assume an exploration variable $\lambda$ controls the extent of exploration within the training distribution $P_{\text{tr}}(G, Y)$. The augmented distribution $\tilde{P}_{\text{tr}}(G, Y)$ is then modelled by the conditional graph distribution $P_{\text{tr}}(G, Y \mid \mathbf{y}_{\text{ood}} = \lambda)$, which can be decomposed as follows:

$$P_{\text{tr}}(G, Y \mid \mathbf{y}_{\text{ood}} = \lambda) \propto p(G) \, p(Y \mid G) \, p(\mathbf{y}_{\text{ood}} = \lambda \mid G, Y) \tag{3}$$

Existing graph generation models (Jo et al., 2022; Martinkus et al., 2022; Vignac et al., 2022) cannot directly sample graphs from the conditional distribution in Equation (3), as it is infeasible to enumerate all possible $\lambda$ values and their corresponding graphs and labels to compute the normalized probabilities. To overcome this limitation, we propose a novel guidance scheme to direct the score-based generative model (Song et al., 2020) towards the target distribution.

**Score-based Graph Generation** $p(G)$ in Equation 3 is the distribution of unlabelled graphs, which can be captured by score-based generative model (Song et al., 2020). The foundational work by (Song et al., 2020) introduced a method for modeling the diffusion process of data into noise and vice versa using stochastic differential equations (SDEs). For graph generation, this diffusion process gradually corrupts graphs into a prior distribution like the normal distribution. The model subsequently samples noise from the prior distribution and learns a score function to denoise the perturbed graphs. Given an unlabeled graph $G$, we use continuous time $t \in [0, T]$ to index the diffusion steps $\{G_t\}_{t=1}^T$ of the graph, where $G_0$ represents the original distribution and $G_T$ follows

Figure 1: Overview of OODA. The diffusion process iteratively transforms an unlabeled original graph $G_0$ into noise $G_T$. During the denoising process, $G_{t-1}$ is computed using the conditional score $\nabla_{G_t} \log p_t (G_t \mid \mathbf{y}_G, \mathbf{y}_{\text{ood}} = \lambda)$, constrained by the target class $\mathbf{y}_G$ and the exploration parameter $\lambda$. Ultimately, the clean OOD graph $\tilde{G}_0$ is generated.

a prior distribution. The forward diffusion process from the graph to the prior distribution is defined through an Itô SDE:

$$\mathrm{d}G_t = \mathbf{f}_t (G_t) \, \mathrm{d}t + g_t \, \mathrm{d}\mathbf{w}, \tag{4}$$

that incorporates linear drift coefficient $\mathbf{f}_t(\cdot) : \mathcal{G} \to \mathcal{G}^1$ and scalar diffusion coefficient $g_t : \mathcal{G} \to \mathbb{R}$ related to the amount of noise corrupting the unlabelled graph at each infinitesimal step $t$, along with a standard Wiener process $\mathbf{w}$. In contrast, the reverse diffusion employs SDE that factors in the gradient fields or scores of the perturbed graphs $G_t$ for denoising and graph generation from $T$ to 0:

$$\mathrm{d}G_t = \left[ \mathbf{f}_t (G_t) - g_t^2 \nabla_{G_t} \log p_t (G_t) \right] \mathrm{d}\bar{t} + g_t \, \mathrm{d}\overline{\mathbf{w}},$$

where $p_t (G_t)$ denotes the marginal distribution at time $t$ in forward diffusion, with $\mathbf{f}_t (G_t)$ and $g_t$ representing the drift and diffusion coefficients, respectively. $\mathrm{d}\overline{\mathbf{w}}$ here is the reverse-time standard Wiener process, and $\mathrm{d}\bar{t}$ is an infinitesimal negative time step. The score network $\boldsymbol{s}_{\theta,t} (G_t)$ is trained to approximate the unknown score function $\nabla_{G_t} \log p_t (G_t)$. Although the score-based generative model can capture the distribution of unlabeled graphs, it cannot generate the pairs $(G, Y)$ from the OOD distribution. To address this limitation, we introduce a novel OOD guidance scheme designed to generate OOD graphs and their corresponding label from a score-based generative model *trained on unlabeled graphs*.

**Score-based Graph Generation with OOD Control** To explore the training distribution in a controlled manner, we propose a novel OOD-controlled score-based graph generative model capable of generating OOD graph samples and their corresponding labels. The extent of exploration in the generative process is regulated by the hyperparameter $\lambda$. An overview is provided in Figure 1. Our approach involves sampling $(G, Y)$ from $P_{\text{tr}} (G, Y \mid \mathbf{y}_{\text{ood}} = \lambda)$ and solving the conditional reverse-time SDE:

$$\mathrm{d}G_t = \left[ \mathbf{f}_t (G_t) - g_t^2 \nabla_{G_t} \log p_t (G_t, \mathbf{y}_G \mid \mathbf{y}_{\text{ood}} = \lambda) \right] \mathrm{d}\bar{t} + g_t \, \mathrm{d}\overline{\mathbf{w}} \tag{5}$$

Where $\mathbf{y}_G$ denotes the graph's label and $\mathbf{y}_{\text{ood}}$ specifies the amount of OOD exploration. To sample explored graph instances from $P_{\text{tr}} (G, Y \mid \mathbf{y}_{\text{ood}} = \lambda)$ using a diffusion model, we note that $\nabla_{G_t} \log p_t (G_t, \mathbf{y}_G \mid \mathbf{y}_{\text{ood}} = \lambda) = \nabla_{G_t} \log p_t (G_t \mid \mathbf{y}_G, \mathbf{y}_{\text{ood}} = \lambda)$. The proof is in Appendix A.1. Therefore, the desired conditional reverse-time SDE becomes:

$$\mathrm{d}G_t = \left[ \mathbf{f}_t (G_t) - g_t^2 \nabla_{G_t} \log p_t (G_t \mid \mathbf{y}_G, \mathbf{y}_{\text{ood}} = \lambda) \right] \mathrm{d}\bar{t} + g_t \, \mathrm{d}\overline{\mathbf{w}} \tag{6}$$

According to Equation 3, the conditional score function $\nabla_{G_t} \log p_t (G_t \mid \mathbf{y}_G, \mathbf{y}_{\text{ood}} = \lambda)$ is the sum of three components:

$$
\begin{aligned}
\nabla_{G_t} \log p_t (G_t \mid \mathbf{y}_G, \mathbf{y}_{\text{ood}} = \lambda) = & \nabla_{G_t} \log p_t (G_t) \\
& + \nabla_{G_t} \log p_t (\mathbf{y}_G \mid G_t) \\
& + \nabla_{G_t} \log p_t (\mathbf{y}_{\text{ood}} = \lambda \mid G_t, \mathbf{y}_G)
\end{aligned}
\tag{7}
$$

Based on Bazhenov et al. (2022); Wu et al. (2023), OOD graphs are those with low likelihood under the original distribution $P_{\text{tr}}(G, Y)$. For instance, in the Motif training dataset (Gui et al., 2022), a house motif (stable subgraph) is only connected with wheel graphs, tree graphs or ladder graphs (environmental subgraphs). Consequently, when environmental graphs are explored, the entire graph patterns exist in the low-density region of the original distribution. Inspired by Lee et al. (2023), we

model the distribution $p_t\left(\mathbf{y}_{\text{ood}} = \lambda \mid G_t, \mathbf{y}_G\right)$ as proportional to the negative exponent of the joint density of $G_t$ and $\mathbf{y}_G$, $p_t\left(G_t, \mathbf{y}_G\right)$:

$$p_t\left(\mathbf{y}_{\text{ood}} = \lambda \mid G_t, \mathbf{y}_G\right) \propto p_t\left(G_t, \mathbf{y}_G\right)^{-\sqrt{\lambda}} = p_t\left(G_t\right)^{-\sqrt{\lambda}} p_t\left(\mathbf{y}_G \mid G_t\right)^{-\sqrt{\lambda}} \tag{8}$$

Accordingly, the gradient of the log probability for conditional reverse diffusion is expressed as:

$$\nabla_{G_t} \log p_t\left(G_t \mid \mathbf{y}_G, \mathbf{y}_{\text{ood}} = \lambda\right) = (1 - \sqrt{\lambda})\nabla_{G_t} \log p_t\left(G_t\right) + (1 - \sqrt{\lambda})\nabla_{G_t} \log p_t\left(\mathbf{y}_G \mid G_t\right) \tag{9}$$

As seen in Equation 9, we need to compute the target class probability for the conditional score functions. To achieve this, we train a classifier $\phi_t$ to predict graph label $\mathbf{y}_G$ from the noisy graph $G_t$ at time step $t$: $\phi_t(G_t) = \hat{\mathbf{y}}_G$. The output probability of $\phi_t$ can approximate the distribution $p_t\left(\mathbf{y}_G \mid G_t\right)$.

Consequently, the conditional score function in Equation 6 results in a marginal distribution proportional to $p_t\left(G_t, \mathbf{y}_G\right)^{1-\sqrt{\lambda}}$. When $\lambda = 0$, the marginal distribution in Equation 6 simplifies to $p_t\left(G_t, \mathbf{y}_G\right)$. The reverse-time diffusion process denoises the perturbed graphs to the augmented distribution $P_{\text{tr}}\left(G, Y \mid \mathbf{y}_{\text{ood}} = 0\right)$, which closely resembles the original distribution $P_{\text{tr}}\left(G, Y\right)$. As $\lambda$ increases, the augmented distribution $P_{\text{tr}}\left(G, Y \mid \mathbf{y}_{\text{ood}} = \lambda\right)$ becomes broader relative to the original data distribution. By adjusting $\lambda$, we can flexibly control the dispersion, making the generated graphs more likely to be out-of-distribution. Consequently, we can increase the divergence between $P_{\text{tr}}\left(G, Y \mid \mathbf{y}_{\text{ood}} = \lambda\right)$ and $P_{\text{tr}}\left(G, Y\right)$.

**Working principles of OODA.** OODA not only explores controllable environmental features but also preserves stable features. According to Equation 7, the term $\nabla_{G_t} \log p_t\left(\mathbf{y}_{\text{ood}} = \lambda \mid G_t, \mathbf{y}_G\right)$ directs the reverse diffusion process towards out-of-distribution regions with graphs that exhibit the explored environmental patterns. Additionally, the term $\nabla_{G_t} \log p_t\left(\mathbf{y}_G \mid G_t\right)$ guides the reverse diffusion process towards regions containing graphs that are highly likely to possess patterns determining the target class. Although we use a classifier to approximate the target class probability $p_t\left(\mathbf{y}_G \mid G_t\right)$, the classifier is able to predict the target class even in the presence of noise. By iteratively adding noise, the classifier captures the relationship between the stable patterns in the perturbed graph $G_t$ and the target class. Together, these two terms guide the sampling process to generate graphs that contain desired stable patterns and extend beyond the training environments.

In addition to the aforementioned principles, OODA can generate diverse and valid OOD graph instances for the final graph classification problem under distributional shifts. This capability is based on the inherent randomness in the forward processes and the effectiveness of the reverse process in the diffusion model.

**Implementations of OODA.** Directly applying this framework to graphs proved inadequate for capturing the intricate relationships between nodes and edges, essential for accurately learning graph distributions (Jo et al., 2022). To overcome this, we simultaneously models the diffusion processes of node features and adjacency matrices of perturbed graphs $\{G_t = (\boldsymbol{X}_t, \boldsymbol{A}_t)\}_{t=0}^T$ using a set of SDEs for Equation 6:

$$\begin{cases} \mathrm{d}\boldsymbol{X}_t = \Big[\mathbf{f}_{1,t}\left(\boldsymbol{X}_t\right) - (1 - \sqrt{\lambda})g_{1,t}^2 \nabla_{\boldsymbol{X}_t} \log p_t\left(\boldsymbol{X}_t, \boldsymbol{A}_t\right) \\ \qquad - (1 - \sqrt{\lambda})g_{1,t}^2 \nabla_{\boldsymbol{X}_t} \log p_t\left(\mathbf{y}_G \mid \boldsymbol{X}_t, \boldsymbol{A}_t\right)\Big]\mathrm{d}\bar{t} + g_{1,t}\,\mathrm{d}\overline{\mathbf{w}}_1 \\ \mathrm{d}\boldsymbol{A}_t = \Big[\mathbf{f}_{2,t}\left(\boldsymbol{A}_t\right) - (1 - \sqrt{\lambda})g_{2,t}^2 \nabla_{\boldsymbol{A}_t} \log p_t\left(\boldsymbol{X}_t, \boldsymbol{A}_t\right) \\ \qquad - (1 - \sqrt{\lambda})g_{2,t}^2 \nabla_{\boldsymbol{A}_t} \log p_t\left(\mathbf{y}_G \mid \boldsymbol{X}_t, \boldsymbol{A}_t\right)\Big]\mathrm{d}\bar{t} + g_{2,t}\,\mathrm{d}\overline{\mathbf{w}}_2. \end{cases} \tag{10}$$

where $\mathbf{f}_t(\boldsymbol{X}, \boldsymbol{A}) = (\mathbf{f}_{1,t}(\boldsymbol{X}), \mathbf{f}_{2,t}(\boldsymbol{A}))$ and $g_t = (g_{1,t}, g_{2,t})$ representing the drift and diffusion coefficients, respectively. The reverse-time processes are captured by standard Wiener processes $\overline{\mathbf{w}}_1$ and $\overline{\mathbf{w}}_2$, with $\mathrm{d}\bar{t}$ indicating an infinitesimally small negative time step. We train one graph transformer (Dwivedi & Bresson, 2020; Vignac et al., 2022), denoted as $s_{\theta,t} = (s_{\theta_1,t}, s_{\theta_2,t})$ to closely estimate the partial score functions $\nabla_{\boldsymbol{X}_t} \log p_t\left(\boldsymbol{X}_t, \boldsymbol{A}_t\right)$ and $\nabla_{\boldsymbol{A}_t} \log p_t\left(\boldsymbol{X}_t, \boldsymbol{A}_t\right)$, facilitating the backward simulation of the equation to simultaneously generate node features and adjacency matrices of unlabelled graphs. Therefore, the denoising graph transformer is only trained with unlabelled graphs without $\lambda$ values and graph labels. The graph transformer with perturbed node features $X_t$,

adjacency matrices $A_t$ and normalized timestep as input. The timestep $t$ value is treated as a global graph feature, and an embedding layer is used to embed $t$.

We also use a graph transformer model $\phi_t$ with the same architecture to predict the class label of the noisy graphs $G_t = (\boldsymbol{X}_t, \boldsymbol{A}_t)$ at time step $t$. The target class $j$ probability $p_t(\mathbf{y}_G = j \mid \boldsymbol{X}_t, \boldsymbol{A}_t)$ is then given by:

$$p_t(\mathbf{y}_G = j \mid \boldsymbol{X}_t, \boldsymbol{A}_t) = \frac{e^{\phi_t(\boldsymbol{X}_t, \boldsymbol{A}_t)_{[j]}}}{\sum_{j=1}^{M} e^{\phi_t(\boldsymbol{X}_t, \boldsymbol{A}_t)_{[j]}}}$$

Once both the score transformer and the classifier are trained, we use them to compute the conditional partial scores during the sampling process.

We adopt the two popular time-dependent hyperparameters $\alpha_{1,t}$ and $\alpha_{2,t}$ for the target class probability predicted by $\phi_t$. These hyperparameters are defined as follows:

$$
\begin{aligned}
\alpha_{1,t} &= 0.1^t \frac{r_1 \|\boldsymbol{s}_{\theta_1,t}(\boldsymbol{G}_t)\|}{\|\nabla_{\boldsymbol{X}_t} \log p_t(\mathbf{y}_G \mid \boldsymbol{X}_t, \boldsymbol{A}_t)\|} \\
\alpha_{2,t} &= 0.1^t \frac{r_2 \|\boldsymbol{s}_{\theta_2,t}(\boldsymbol{G}_t)\|}{\|\nabla_{\boldsymbol{A}_t} \log p_t(\mathbf{y}_G \mid \boldsymbol{X}_t, \boldsymbol{A}_t)\|}
\end{aligned}
\tag{11}
$$

where $\alpha_t = (\alpha_{1,t}, \alpha_{2,t})$, $r_1$ and $r_2$ are the weights for node features and adjacency matrices respectively, and $\|\cdot\|$ is the entry-wise matrix norm.

Intuitively, at the early stages of the reverse-time SDEs, the graphs are highly perturbed, resembling the prior noise distribution. Therefore, the classifier cannot accurately approximate the target class probability. Consequently, in the initial denoising steps, we focus more on guiding the reverse-time SDEs towards the OOD distribution. As the reverse-time SDEs progressively denoise the graphs, we introduce guidance to direct the reverse-time SDEs towards regions exhibiting the desired stable patterns and the explored OOD environmental patterns.

## 5 EXPERIMENTS

In this section, we first demonstrate the effectiveness of our diffusion models on graph OOD tasks in Section 5.2, then validate the efficacy of our OOD-controlled diffusion process on GOOD-Motif and GOOD-HIV datasets in Section 5.3. We further conduct an ablation study to verify the effectiveness of our diffusion models to generate OOD graphs in Section 5.3

### 5.1 EXPERIMENTAL SETTINGS

**Setup.** For a fair comparison, we adopt the same evaluation metrics as those used in (Gui et al., 2022). The model that achieves the best performance on the OOD validation sets is then evaluated on the OOD test sets. Furthermore, to ensure fair comparison across all methods, we utilize the same GNN backbones—GIN (Xu et al., 2019) and GIN-Virtual (Xu et al., 2018; Gilmer et al., 2017)—as applied in the GOOD benchmark (Gui et al., 2022) for each dataset. The experimental details, including evaluation metrics and hyperparameter configurations, are summarized in Appendix A.4.

**Datasets.** We use synthetic, semi-artificial, and real-world datasets from GOOD (Gui et al., 2022), including GOOD-Motif, GOOD-CMNIST, GOOD-HIV, and GOOD-SST2. Consistent with (Gui et al., 2022), we apply base, size, color, scaffold, and length data splits to introduce diverse covariate shifts in graph structure, node features, and edge features. Detailed descriptions of the datasets are provided in Appendix A.4.

**Baselines.** We adopt 16 baselines, which can be divided into the following three specific categories:(i) *general generalization algorithms*, including ERM, IRM (Arjovsky et al., 2019), Group-DRO (Sagawa et al., 2019), VREx (Krueger et al., 2021), DANN (Ganin et al., 2016), Deep Coral (Sun & Saenko, 2016); (ii) *graph generalization algorithms*, including DIR (Wu et al., 2022), GSAT (Miao et al., 2022), CIGA (Chen et al., 2022), and (iii) *graph data augmentation techniques*, including DropNode (Feng et al., 2020), DropEdge (Rong et al., 2019), MaskFeature (Thakoor et al., 2021), FLAG (Kong et al., 2022), M-Mixup (Wang et al., 2021), G-Mixup (Han et al., 2022), AIA (Sui et al., 2024).

| Type | Method | Motif | | CMNIST | Molhiv | | GOOD-SST2 |
|---|---|---|---|---|---|---|---|
| | | base | size | color | scaffold | size | length |
| General Generalization | ERM | $68.66 \pm 4.25$ | $51.74 \pm 2.88$ | $28.60 \pm 1.87$ | $69.58 \pm 2.51$ | $59.94 \pm 2.37$ | $81.30 \pm 0.35$ |
| | IRM | $70.65 \pm 4.17$ | $51.41 \pm 3.78$ | $27.83 \pm 2.13$ | $67.97 \pm 1.84$ | $59.00 \pm 2.92$ | $79.91 \pm 1.97$ |
| | GroupDRO | $68.24 \pm 8.92$ | $51.95 \pm 5.86$ | $29.07 \pm 3.14$ | $70.64 \pm 2.57$ | $58.98 \pm 2.16$ | $81.35 \pm 0.54$ |
| | VREx | $71.47 \pm 6.69$ | $52.67 \pm 5.54$ | $28.48 \pm 2.87$ | $70.77 \pm 2.84$ | $58.53 \pm 2.88$ | $80.64 \pm 0.35$ |
| | DANN | $65.47 \pm 5.35$ | $51.46 \pm 3.41$ | $29.14 \pm 2.93$ | $70.63 \pm 1.82$ | $\underline{62.38 \pm 2.65}$ | $79.71 \pm 1.35$ |
| | Deep Coral | $68.88 \pm 3.61$ | $53.71 \pm 2.75$ | $29.05 \pm 2.19$ | $68.61 \pm 1.70$ | $60.11 \pm 3.53$ | $79.81 \pm 0.22$ |
| Graph Generalization | DIR | $62.07 \pm 8.75$ | $52.27 \pm 4.56$ | $33.20 \pm 6.17$ | $68.07 \pm 2.29$ | $58.08 \pm 2.31$ | $77.65 \pm 1.93$ |
| | GSAT | $62.80 \pm 11.41$ | $53.20 \pm 8.35$ | $28.17 \pm 1.26$ | $68.66 \pm 1.35$ | $58.06 \pm 1.98$ | $81.49 \pm 0.76$ |
| | CIGA | $66.43 \pm 11.31$ | $49.14 \pm 8.34$ | $32.22 \pm 2.67$ | $69.40 \pm 2.39$ | $59.55 \pm 2.56$ | $80.44 \pm 1.24$ |
| Graph Augmentation | DropNode | $\underline{74.55 \pm 5.56}$ | $54.14 \pm 3.11$ | $33.01 \pm 0.12$ | $\underline{71.18 \pm 1.16}$ | $58.52 \pm 0.49$ | $81.14 \pm 1.73$ |
| | DropEdge | $45.08 \pm 4.46$ | $45.63 \pm 4.61$ | $22.65 \pm 2.90$ | $70.78 \pm 1.38$ | $58.53 \pm 1.26$ | $78.93 \pm 1.34$ |
| | MaskFeature | $64.98 \pm 6.95$ | $52.24 \pm 3.75$ | $44.85 \pm 2.42$ | $65.90 \pm 3.68$ | $62.30 \pm 3.17$ | $\mathbf{82.00 \pm 0.73}$ |
| | FLAG | $61.12 \pm 5.39$ | $51.66 \pm 4.14$ | $32.30 \pm 2.69$ | $68.45 \pm 2.30$ | $60.59 \pm 2.95$ | $77.05 \pm 1.27$ |
| | M-Mixup | $70.08 \pm 3.82$ | $51.48 \pm 4.91$ | $26.47 \pm 3.45$ | $68.88 \pm 2.63$ | $59.03 \pm 3.11$ | $80.88 \pm 0.60$ |
| | G-Mixup | $59.66 \pm 7.03$ | $52.81 \pm 6.73$ | $31.85 \pm 5.82$ | $70.01 \pm 2.52$ | $59.34 \pm 2.43$ | $80.28 \pm 1.49$ |
| | AIA | $73.64 \pm 5.15$ | $\underline{55.85 \pm 7.98}$ | $\underline{36.37 \pm 4.44}$ | $71.15 \pm 1.81$ | $61.64 \pm 3.37$ | $81.69 \pm 0.57$ |
| | OODA(Ours) | $\mathbf{75.25 \pm 3.84}$ | $\mathbf{60.81 \pm 7.80}$ | $\mathbf{54.60 \pm 2.27}$ | $\mathbf{72.67 \pm 1.28}$ | $\mathbf{66.47 \pm 2.29}$ | $\mathbf{82.69 \pm 0.28}$ |

Table 1: Performance on synthetic and real-world datasets. Bold numbers indicate the best performance, while the underlined numbers indicate the second best performance.

## 5.2 GRAPH OUT-OF-DISTRIBUTION CLASSIFICATION

The graph classification performances under covariate shift are presented in Table 1. As shown, OODA consistently outperforms all baseline methods across diverse covariate shifts and different datasets.

On the synthetic dataset GOOD-Motif, OODA achieves a performance improvement of $6.59\%$ over ERM under base shift and $9.07\%$ under size shift. For the semi-artificial dataset GOOD-CMNIST, designed for node feature shifts, performance is significantly enhanced by $18.23\%$ compared to the leading graph augmentation method, AIA, and improved by $21.40\%$ over the best graph invariant learning method, DIR. In the real-world molecular dataset GOOD-HIV, where covariate shifts occur in graph structure, node features, and edge features simultaneously, OODA outperforms ERM by $2.09\%$ on scaffold shift and by $6.53\%$ on size shift. For the real-world natural language sentiment analysis dataset GOOD-SST2, while AIA is outperformed by MaskFeature by $0.31\%$, OODA exceeds MaskFeature by $0.69\%$.

These results demonstrate that no baseline graph invariant learning or graph data augmentation methods consistently outperform each other under various covariate shifts. OODA enhances environmental exploration by generating out-of-distribution (OOD) graphs in a controlled manner while preserving stable features. Consequently, OODA reliably improves performance across different datasets facing various covariate shifts.

## 5.3 OOD CONTROLLED GRAPH GENERATION

In this section, we present both qualitative and quantitative experiments to demonstrate the effectiveness of our out-of-distribution diffusion augmentation framework. The experiments are conducted using the GOOD-Motif-base and GOOD-HIV-scaffold datasets.

**Controlled OOD generation.** We first validate that our framework can explore the space of the original distribution $P_{\mathrm{tr}}(G, Y)$ and generate an augmented distribution $\tilde{P}_{\mathrm{tr}}(G, Y)$ in a controlled manner. To evaluate the deviation of the augmented distribution from the training graphs, we employ existing random Graph Isomorphism Networks (GIN)-based metrics (Thompson et al., 2022), which are well-suited for graph generative models. These metrics are more expressive, and significantly reduce computational costs, particularly on large graph datasets. As recommended by (Thompson et al., 2022), we utilize the maximum mean discrepancy (MMD) with a radial basis function (RBF) kernel, which is widely recognized as a robust metric for measuring distributional differences. MMD RBF is used to measure the discrepancy between the aug-

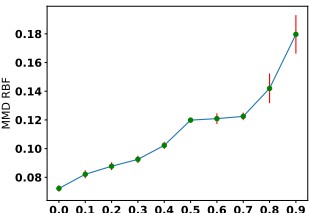

Figure 2: Distance between original and OOD graph distributions on GOOD-Motif.

mented and training distributions. We vary $\lambda$ uniformly in the range $[0, 1)$ with a step size of $0.1$

to generate ten augmented datasets, each containing the same number of graphs as the training set, and compute the MMD RBF between each augmented dataset and the original training dataset. A detailed description of the estimation procedure is provided in Appendix A.3. The results for GOOD-Motif with base covariate shift are presented in Figure 2. As shown in Figure 2, the MMD RBF between the original GOOD-Motif-base graph distribution $P_{\text{tr}}(G, Y)$ and the augmented graph distribution $\tilde{P}_{\text{tr}}(G, Y)$ increases as $\lambda$ grows. When $\lambda = 0.0$, the MMD RBF is $0.072 \pm 0.002$. As $\lambda$ increases to $0.9$, the MMD RBF becomes $1.5$ times larger than that at $\lambda = 0.0$. These results demonstrate that OODA can generate OOD graph samples in a controlled manner by flexibly adjusting the value of $\lambda$.

**Stable patterns preservation.** we utilize a pretrained graph transformer $\phi_t$, which was trained on noisy graphs from the training distribution, to compute the probability of each augmented OOD graph sample possessing stable patterns that determine the target class, denoted as $p\left(\mathbf{y}_G \mid \tilde{G}_0\right)$, where $\mathbf{y}_G$ represents the target class of $\tilde{G}_0$. We then compute the expected value $\mathbb{E}_{\tilde{G}_0 \sim \tilde{P}}\left[p\left(\mathbf{y}_G \mid \tilde{G}_0\right)\right]$ across the augmented distribution. This approach verifies that the explored graphs in $\tilde{P}_{\text{tr}}(G, Y)$ preserve the stable patterns characteristic of the graphs in $P_{\text{tr}}(G, Y)$. The results for GOOD-Motif with base covariate shifts are illustrated in Figure 3. As depicted, the expected probability $\mathbb{E}_{\tilde{G}_0 \sim \tilde{P}}\left[p\left(\mathbf{y}_G \mid \tilde{G}_0\right)\right]$ consistently exceeds $0.95$ as $\lambda$ increases, demonstrating the capability of OODA to generate OOD graphs that retain stable patterns.

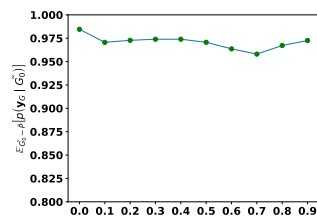

Figure 3: Expected probabilities that the OOD GOOD-Motif graphs retain stable patterns.

**Visualization of OOD graphs.** We further demonstrate the efficacy of OODA by visualizing the OOD graphs generated by our approach in Table 3. In this visualization, three label-determining motifs—house, cycle, and crane—are highlighted in red, while the three environmental base graphs in the training distribution—wheel, tree, and ladder—are indicated in green. As illustrated in Table 3, increasing $\lambda$ leads to gradual modifications in the structures of the base graphs, while the motifs remain preserved.

**OOD molecules generation.** To further validate the effectiveness of OODA in generating valid OOD molecules, we assess both the exploration of new patterns and the preservation of stable features. The Fréchet ChemNet Distance (FCD) (Preuer et al., 2018) is employed to quantify the distance between the training and augmented distributions of molecules, based on the penultimate activations of ChemNet. Additionally, RDKit (Landrum et al., 2016) is used to evaluate the fraction of valid molecules. We also compute the $\mathbb{E}_{\tilde{G}_0 \sim \tilde{P}}\left[p\left(\mathbf{y}_G \mid \tilde{G}_0\right)\right]$ to confirm that the OOD molecules retain stable patterns necessary for inhibiting HIV replica-

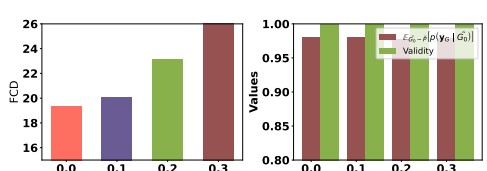

Figure 4: (Left): Distance between the original GOOD-HIV graph distribution and the augmented graph distribution. (Right): The validity of the OOD molecules and the expected probabilities that the OOD GOOD-HIV molecules retain stable patterns.

tion. The results, shown in Figure 4, demonstrate that as $\lambda$ increases, the FCD between the training molecules and the generated OOD molecules grows. Despite this, the OOD molecules consistently preserve stable patterns, with $\mathbb{E}_{\tilde{G}_0 \sim \tilde{P}}\left[p\left(\mathbf{y}_G \mid \tilde{G}_0\right)\right]$ remaining above $0.97$, while maintaining a $100\%$ validity rate.

**Ablation study.** In this section, we present experiments to evaluate the impact of exploration guidance ($\lambda$) and stable patterns preservation guidance ($\alpha$). The results are summarized in Table 2. As shown, incorporating $\alpha$ guidance improves the OOD performance of the diffusion model trained on unlabeled graphs by ensuring that the generated graphs retain the stable patterns that determine the labels. Without $\alpha$ guidance, a diffusion model guided only by $\lambda$ tends to push the augmented graphs into arbitrary out-of-distribution (OOD) regions, which negatively impacts performance to

| $\lambda$ | $\alpha$ | Motif-base | Molhiv-scaffold | GOOD-SST2-length |
|---|---|---|---|---|
| ERM | | $68.66 \pm 4.25$ | $69.58 \pm 2.51$ | $81.30 \pm 0.35$ |
| ✗ | ✗ | $68.55 \pm 6.04$ | $68.94 \pm 1.26$ | $80.86 \pm 0.76$ |
| ✓ | ✗ | $66.25 \pm 7.42$ | $70.01 \pm 1.71$ | $78.87 \pm 3.04$ |
| ✗ | ✓ | $74.57 \pm 4.50$ | $71.71 \pm 1.77$ | $81.59 \pm 0.65$ |
| OODA | | $\mathbf{75.25 \pm 3.84}$ | $\mathbf{72.67 \pm 1.28}$ | $\mathbf{82.69 \pm 0.28}$ |

Table 2: Performance of OODA w/o environmental exploration guidance and stable pattern preservation guidance on synthetic and real-world datasets. Bold numbers indicate the best performance.

Table 3: Visualizations of the augmented GOOD-Motif-base graphs generated by OODA. The graphs $\tilde{G}$ with $\lambda = 0.1$, $\lambda = 0.2$, and $\lambda = 0.3$ represent the OOD graphs generated by OODA under different values of $\lambda$.

some extent. Ultimately, the combination of both $\alpha$ and $\lambda$ guidance enables the augmented distribution to capture both stable patterns and novel environmental patterns, resulting in the best overall performance.

## 6 CONCLUSION

In this work, we proposed OODA, an out-of-distribution graph generation framework based on a score-based diffusion probabilistic model, designed to address covariate shifts in graph learning. Our approach generates OOD graph samples that integrate both explored environmental and stable features, eliminating the need to separate them during training. Furthermore, OODA can simultaneously explore new environments in graph structure, node features, and edge features. While score-based diffusion models demonstrate significant potential in handling diverse covariate shifts, they present scalability challenges when applied to large-scale graphs. Additionally, generating OOD graphs may require careful tuning of hyperparameters in the guidance scheme to balance exploration quality, particularly across different datasets. In this study, we applied minimal hyperparameter tuning to achieve competitive results. Future work could focus on developing scalable diffusion models and exploring parameter-efficient tuning strategies to further enhance OOD graph generation.

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

# A APPENDIX

## A.1 PROOFS

$$\nabla_{G_t} \log p_t \left( G_t, \mathbf{y}_G \mid \mathbf{y}_{\text{ood}} = \lambda \right) = \nabla_{G_t} \log p_t \left( G_t \mid \mathbf{y}_G, \mathbf{y}_{\text{ood}} = \lambda \right)$$

Proof:
$$\log p_t \left( G_t, \mathbf{y}_G \mid \mathbf{y}_{\text{ood}} = \lambda \right) = \log p_t \left( G_t, \mathbf{y}_G, \mathbf{y}_{\text{ood}} = \lambda \right) - \log p_t \left( \mathbf{y}_{\text{ood}} = \lambda \right)$$

Since $p_t \left( \mathbf{y}_{\text{ood}} = \lambda \right)$ is independent of $G_t$, $\nabla_{G_t} \log p_t \left( \mathbf{y}_{\text{ood}} = \lambda \right) = 0$. Therefore,

$$\nabla_{G_t} \log p_t \left( G_t, \mathbf{y}_G \mid \mathbf{y}_{\text{ood}} = \lambda \right) = \nabla_{G_t} \log p_t \left( G_t, \mathbf{y}_G, \mathbf{y}_{\text{ood}} = \lambda \right)$$

Additionally,
$$\log p_t \left( G_t \mid \mathbf{y}_G, \mathbf{y}_{\text{ood}} = \lambda \right) = \log p_t \left( G_t, \mathbf{y}_G, \mathbf{y}_{\text{ood}} = \lambda \right) - \log p_t \left( \mathbf{y}_G, \mathbf{y}_{\text{ood}} = \lambda \right)$$

Since $p_t \left( \mathbf{y}_G, \mathbf{y}_{\text{ood}} = \lambda \right)$ is independent of $G_t$, $\nabla_{G_t} \log p_t \left( \mathbf{y}_G, \mathbf{y}_{\text{ood}} = \lambda \right) = 0$. Therefore,

$$\nabla_{G_t} \log p_t \left( G_t \mid \mathbf{y}_G, \mathbf{y}_{\text{ood}} = \lambda \right) = \nabla_{G_t} \log p_t \left( G_t, \mathbf{y}_G, \mathbf{y}_{\text{ood}} = \lambda \right)$$

Finally,
$$\nabla_{G_t} \log p_t \left( G_t, \mathbf{y}_G \mid \mathbf{y}_{\text{ood}} = \lambda \right) = \nabla_{G_t} \log p_t \left( G_t \mid \mathbf{y}_G, \mathbf{y}_{\text{ood}} = \lambda \right)$$

## A.2 GRAPH COVARIATE SHIFT

In this work, we address the challenge of OOD graph classification, where the goal is to develop models trained on a dataset $\mathcal{D}_{\text{tr}}$ that can generalize effectively to a dataset $\mathcal{D}_{\text{te}}$. Considering invariant perspectives under covariate shifts (Gui et al., 2022; Sui et al., 2024), we note while $P_{\text{tr}}(Y \mid G) = P_{\text{te}}(Y \mid G)$ holds, the marginal distributions of graphs differ, i.e., $P_{\text{tr}}(G) \neq P_{\text{te}}(G)$. This discrepancy may stem from the training set's limited size or diversity and the unpredictable conditions in test environments. For instance, acquiring sufficient samples of molecules for property prediction can be costly and challenging. Moreover, geometric deep learning models are frequently applied to predict properties of molecules with unseen scaffolds. Prior study (Li et al., 2024) identifies two primary types of covariate distribution discrepancies: (1) $P_{\text{tr}}(\boldsymbol{X}) \neq P_{\text{te}}(\boldsymbol{X})$ while $P_{\text{tr}}(\boldsymbol{A}) = P_{\text{te}}(\boldsymbol{A})$ and $P_{\text{tr}}(\boldsymbol{E}) = P_{\text{te}}(\boldsymbol{E})$, exemplified by the GOOD-CMNIST dataset (Gui et al., 2022), where digits of different colors indicate different environments that correspond to dataset splits. (2) $P_{\text{tr}}(\boldsymbol{A}, \boldsymbol{E}) \neq P_{\text{te}}(\boldsymbol{A}, \boldsymbol{E})$ or $P_{\text{tr}}(\boldsymbol{A}, \boldsymbol{X}, \boldsymbol{E}) \neq P_{\text{te}}(\boldsymbol{A}, \boldsymbol{X}, \boldsymbol{E})$, as seen in the GOOD-Motif dataset (Gui et al., 2022) where training and testing environments differ in graph size and bases. In the GOOD-HIV benchmark (Gui et al., 2022), the molecular scaffolds differ between the training and testing datasets.

The covariate shift, whether in graph structure, node features or eddge features, poses significant challenges to OOD generalization. This work aims to provide a general framework to address all three types of covariate shifts.

## A.3 METRICS FOR MEASURING DISTRIBUTIONAL DIFFERENCES

In this section, we provide detailed implementation steps for measuring the distributional differences between the augmented dataset and the training dataset. Following (Thompson et al., 2022), we first use an untrained random GIN, $h$, to extract graph embeddings from both the augmentation distribution and the training distribution. The maximum mean discrepancy (MMD) is then computed to quantify the dissimilarity between the graph embedding distributions:

$$\text{MMD}^2(P \| \tilde{P}) = \mathbb{E}_{g, \tilde{g} \sim P}[k(h(g), h(\tilde{g}))] + \mathbb{E}_{g, \tilde{g} \sim \tilde{P}}[k(h(g), h(\tilde{g}))] - 2\mathbb{E}_{g \sim P, \tilde{g} \sim \tilde{P}}[k(h(g), h(\tilde{g}))]$$

where $k(\cdot, \cdot)$ is the RBF kernel proposed by (You et al., 2018). As recommended by (Thompson et al., 2022), the MMD RBF scalar is also one of the most reliable metrics for measuring distributional differences:

$$k\left(h(g), h(\tilde{g})\right) = \exp\left(-d\left(h(g), h(\tilde{g})\right)/2\sigma^2\right)$$

Additionally, we employ the Earth Mover's Distance (EMD) from (Thompson et al., 2022) to compute pairwise distances $d(\cdot, \cdot)$.

## A.4 Experimental Details

### A.4.1 Dataset Details

We utilize six datasets from the GOOD benchmark (Gui et al., 2022), including GOOD-Motif-base, GOOD-Motif-size, GOOD-CMNIST-color, GOOD-HIV-scaffold, GOOD-HIV-size, and GOOD-SST2-length. The GOOD benchmark (Gui et al., 2022) is the state-of-the-art framework for systematically evaluating graph OOD generalization. It carefully designs data environments to induce reliable and valid distribution shifts. The selected datasets span a diverse range of domains, covering covariate shifts in general graphs, image-transformed graphs, molecular graphs, and natural language sentiment analysis graphs. The dataset details are as follows:

- **GOOD-Motif:** GOOD-Motif is a synthetic dataset from Spurious-Motif (Wu et al., 2022) specifically designed to investigate structure shifts. Each graph consists of an environmental base graph connected to a label-determining motif. The two primary covariate shift domains are the base graph type and graph size. For base covariate shift, the training distribution includes graphs with wheel, tree, and ladder base structures, while the validation set features star base graphs, and the test set contains path base graphs. For size covariate shift, the training distribution consists of graphs with sizes ranging from 6 to 45 nodes, the validation set contains graphs with sizes between 20 and 75 nodes, and the test set comprises graphs with sizes ranging from 68 to 155 nodes.

- **GOOD-CMNIST:** GOOD-CMNIST is a semi-synthetic dataset designed to investigate node feature shifts. It consists of graphs transformed from MNIST handwritten digit images using superpixel techniques (Monti et al., 2017). Node color features are manually applied, making the color shift environment independent of the underlying structure. Specifically, for covariate shift, digits are colored using seven different colors. The training distribution includes digits colored with the first five colors, while the validation and test distributions contain digits with the remaining two colors, respectively.

- **GOOD-HIV:** GOOD-HIV is a small-scale, real-world molecular dataset sourced from MoleculeNet (Wu et al., 2018). The nodes in these molecular graphs represent atoms, and the edges represent chemical bonds. This dataset is designed to study node feature shifts, edge feature shifts, and structure shifts. The two covariate shift domains are scaffold graph type and molecular graph size. For the scaffold covariate shift, environments are partitioned based on the Bemis-Murcko scaffold (Bemis & Murcko, 1996), a two-dimensional structural base that does not determine a molecule's ability to inhibit HIV replication. For the size covariate shift, the training distribution consists of molecular graphs ranging in size from 17 to 222 atoms. The validation set contains molecules with sizes between 15 and 16 atoms, while the test set includes molecules with sizes from 2 to 14 atoms.

- **GOOD-SST2:** GOOD-SST2 is a real-world natural language sentimental analysis dataset from (Yuan et al., 2022), designed to investigate node feature shifts and structure shifts. Each graph is derived from a sentence, transformed into a grammar tree, where nodes represent words, and node features are corresponding word embeddings. The task is to predict the sentiment polarity of each sentence. Sentence length is chosen as the covariate shift environment, as sentence length should not inherently affect sentiment polarity. For the length covariate shift, the training distribution consists of grammar graphs with sizes ranging from 1 to 7 nodes, the validation distribution includes graphs with sizes from 8 to 14 nodes, and the test distribution contains graphs with sizes from 15 to 56 nodes.

### A.4.2 Implementation settings

**Diffusion models:** Following (Jo et al., 2022), we preprocess each graph into two matrices: $\boldsymbol{X} \in \mathbb{R}^{n \times a}$ for node features, and $\boldsymbol{A} \in \mathbb{R}^{n \times n \times b}$ for adjacency and edge features. Here, $n$ represents the

Table 4: Hyperparameters of diffusion models.

| | Hyperparameter | Motif | CMNIST | Molhiv | GOOD-SST2 |
|---|---|---|---|---|---|
| $s_\theta$ | Number of graph transformer layers | 8 | 8 | 9 | 8 |
| | Number of attention heads | 8 | 8 | 8 | 8 |
| | Hidden dimension of $\boldsymbol{X}$ | 256 | 256 | 256 | 256 |
| | Hidden dimension of $\boldsymbol{A}$ | 64 | 64 | 64 | 64 |
| SDE for $\boldsymbol{X}$ | Type | VP | VP | VP | VP |
| | Number of sampling steps | 1000 | 1000 | 1000 | 1000 |
| | $\beta_{\min}$ | 0.1 | 0.1 | 0.1 | 0.1 |
| | $\beta_{\max}$ | 1.0 | 1.0 | 1.0 | 1.0 |
| SDE for $\boldsymbol{A}$ | Type | VP | VP | VE | VP |
| | Number of sampling steps | 1000 | 1000 | 1000 | 1000 |
| | $\beta_{\min}$ | 0.1 | 0.1 | 0.2 | 0.2 |
| | $\beta_{\max}$ | 1.0 | 1.0 | 1.0 | 0.8 |
| Solver | Type | EM + Langevin | EM + Langevin | Reverse | EM |
| | SNR | 0.2 | 0.2 | 0.0 | 0.0 |
| | Scale coefficient | 0.7 | 0.7 | 0.0 | 0.0 |
| Train | Optimizer | AdamW | AdamW | AdamW | AdamW |
| | Learning rate | $4 \times 10^{-4}$ | $4 \times 10^{-4}$ | $2 \times 10^{-4}$ | $2 \times 10^{-4}$ |
| | Weight decay | $1 \times 10^{-12}$ | $1 \times 10^{-12}$ | $1 \times 10^{-12}$ | $1 \times 10^{-12}$ |
| | Batch size | 128 | 64 | 512 | 64 |
| | EMA | 0.999 | 0.999 | 0.999 | 0.999 |

maximum number of nodes in a graph for the given dataset, while $a$ and $b$ denote the dimensions of node features and edge features, respectively. The graph structure, including edge features, is encoded in $\boldsymbol{A}$. For the GOOD-Motif dataset, $a$ corresponds to the node degree of a node. In GOOD-CMNIST, each node feature is the concatenation of its degree and color. In GOOD-SST2, the node feature is the word embedding. In the molecular dataset GOOD-HIV, $a$ represents possible atom types and $b$ denotes the types of bonds (e.g., single, double, triple). All molecules are converted to their kekulized form, with hydrogens removed using the RDKit library (Landrum et al., 2016). Additionally, we apply the valency correction proposed by (Zang & Wang, 2020) to post-process the generated molecules.

We train a graph transformer model (Dwivedi & Bresson, 2020; Vignac et al., 2022), $s_{\theta,t}$, to approximate the partial score functions $\nabla_{\boldsymbol{X}_t} \log p_t (\boldsymbol{X}_t, \boldsymbol{A}t)$ and $\nabla_{\boldsymbol{A}_t} \log p_t (\boldsymbol{X}_t, \boldsymbol{A}_t)$ for the unlabelled graphs in the OOD training set and evaluate them on the OOD validation set. In line with (Jo et al., 2022), we use VP or VE SDEs to model the diffusion process for both node features and adjacency matrices. The specific details of the diffusion models are provided in Table 4.

We also train a graph transformer model, $\phi_t$, with the same architecture described in Table 4, to predict the class labels of the noisy graphs $G_t = (\boldsymbol{X}_t, \boldsymbol{A}_t)$ at each time step $t$.

**Graph Out-of-Distribution Classification**: Following prior work (Gui et al., 2022; Li et al., 2024), we employ GIN-Virtual (Xu et al., 2018; Gilmer et al., 2017) as the GNN backbone for the GOOD-CMNIST, GOOD-HIV, and GOOD-SST2 datasets. For the GOOD-Motif dataset, we adopt GIN (Xu et al., 2019). To ensure a fair comparison across all methods, we utilize the same GNN backbone architecture for all models.

For each experiment, we select the best checkpoints for OOD testing based on the performance on the OOD validation sets. All experiments are optimized using the Adam optimizer, with weight decay selected from $\{0, 1 \times 10^{-2}, 1 \times 10^{-3}, 1 \times 10^{-4}\}$ and a dropout rate of 0.5. The number of convolutional layers in the GNN models is tuned from the set $\{3, 5\}$, with mean global pooling and ReLU activation. The hidden layer dimension is set to 300. We explore the maximum number of epochs from $\{100, 200, 500\}$, the initial learning rate from $\{1 \times 10^{-3}, 3 \times 10^{-3}, 5 \times 10^{-3}, 1 \times 10^{-4}\}$, and the batch size from $\{32, 64, 128\}$. All models are trained to convergence.

For computation, we typically run each experiment on an NVIDIA GeForce RTX 4090. We report results as the mean and standard deviation across 10 random runs for all experiments.

We perform a grid search for the hyperparameter $\alpha \in \{0.5, 1.0\}$ across all datasets and find that $\alpha = 0.5$ consistently yields satisfactory results throughout the experiments. For $\lambda$, the grid search is tailored to each dataset. Specifically, we explore $\lambda \in \{0.01, 0.02, 0.03, 0.04, 0.05\}$ for the GOOD-Motif-base and GOOD-HIV-scaffold datasets. For GOOD-CMNIST-color, we tune

$\lambda \in \{0.05, 0.1\}$. In the case of GOOD-SST2-length, where $\lambda = 0.01$ corresponds to an increase of one node in the graph size relative to the training distribution, we expand the grid search to $\lambda \in \{0.01, 0.02, ..., 0.14\}$. Similarly, for GOOD-Motif-size, where $\lambda = 0.01$ reflects an increase of one node, we use a search space of $\lambda \in \{0.01, 0.02, 0.03, 0.04, 0.05\}$. For GOOD-HIV-size, where $\lambda = 0.01$ corresponds to a decrease of ten nodes in graph size from the training distribution, we also use $\lambda \in \{0.01, 0.02, 0.03, 0.04, 0.05\}$. Since this hyperparameter tuning is performed during the sampling phase rather than the training phase, it is not computationally intensive.

## A.5 BASELINE SETTINGS

The implementation details for GNN backbones and hyperparameter tuning are consistent with those outlined in Appendix A.4.2. For methods including ERM, IRM (Arjovsky et al., 2019), GroupDRO (Sagawa et al., 2019), VREx (Krueger et al., 2021), DANN (Ganin et al., 2016), Deep Coral (Sun & Saenko, 2016), DIR (Wu et al., 2022), DropNode (Feng et al., 2020), DropEdge (Rong et al., 2019), MaskFeature (Thakoor et al., 2021), FLAG (Kong et al., 2022), M-Mixup (Wang et al., 2021), and G-Mixup (Han et al., 2022), we report results from the study in (Li et al., 2024), which uses the same GNN backbones and hyperparameter tuning as specified in Appendix A.4.2. For GSAT (Miao et al., 2022), CIGA (Chen et al., 2022), and AIA (Sui et al., 2024), we use their publicly available source code, adopting default settings and hyperparameters as detailed in their papers.

## A.6 ADDITIONAL EXPERIMENTAL RESULTS

We also utilize the GOOD-SST2-length dataset to validate that OODA can generate an augmented distribution, $\tilde{P}_{\mathrm{tr}}(G, Y)$, in a controlled manner while preserving stable patterns. In the context of GOOD-SST2-length, the parameter $\lambda$ regulates the OOD size of the augmented graphs, with $\lambda = 0.1$ corresponding to an increase of one unit in the graph size from the training distribution. We systematically vary $\lambda$ within the range $[0, 1)$ in increments of $0.1$ to generate ten augmented datasets, each containing the same number of graphs as the training set. The results for GOOD-SST2-length under length covariate shift are illustrated in Figure 5. As shown in Figure 5 (Left), the MMD RBF between the original GOOD-SST2-length graph distribution, $P_{\mathrm{tr}}(G, Y)$, and the augmented graph distribution, $\tilde{P}_{\mathrm{tr}}(G, Y)$, increases with increasing values of $\lambda$. Specifically, when $\lambda = 0.0$, the MMD RBF is $0.032 \pm 0.005$. As $\lambda$ rises to $0.9$, the MMD RBF escalates to $25$ times its value at $\lambda = 0.0$. These results indicate that OODA can effectively generate OOD graph samples in a controlled manner by flexibly adjusting $\lambda$.

Moreover, as depicted in Figure 5 (Right), the expected probability $\mathbb{E}_{\tilde{G}_0 \sim \tilde{P}} \left[ p \left( \mathbf{y}_G \mid \tilde{G}_0 \right) \right]$ of the augmented GOOD-SST2-length graph distribution consistently exceeds $0.94$ as $\lambda$ increases. This result demonstrates the capability of OODA to generate OOD graphs that retain stable patterns.

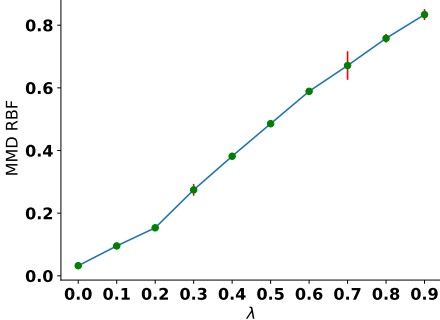 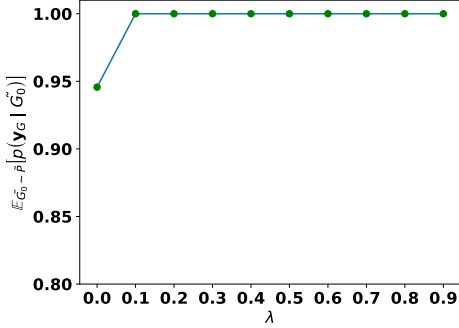

Figure 5: (Left): Distance between the original GOOD-SST2-length graph distribution $P_{\mathrm{tr}}(G, Y)$ and the augmented graph distribution $\tilde{P}_{\mathrm{tr}}(G, Y)$. (Right): Expected probabilities that the augmented GOOD-SST2-length graph distributions retain stable patterns that determine the target class.

Table 5: Performance on the GOOD-Motif-base and GOOD-HIV-scaffold datasets across varying $\lambda$ values.

| $\lambda$ | GOOD-Motif-base | GOOD-HIV-scaffold |
|---|---|---|
| 0.01 | 91.80 | 78.45 |
| 0.02 | 92.97 | 78.50 |
| 0.03 | 92.83 | 79.49 |
| 0.04 | 93.03 | 78.71 |
| 0.05 | 92.77 | 78.69 |

Table 6: Performance on the GOOD-CMNIST-color dataset across varying $\lambda$ values.

| $\lambda$ | GOOD-CMNIST-color |
|---|---|
| 0.05 | 68.66 |
| 0.1 | 67.91 |

## A.7 SENSITIVITY OF HYPERPARAMETER $\lambda$

We determine the value of $\lambda$ by evaluating its effectiveness on OOD validation sets across various datasets. The sensitivity of our method to different $\lambda$ values is illustrated in Tables 5 and 6.

## A.8 TIME AND MEMORY COMPLEXITY

Our pipeline consists of two stages: data augmentation, and trainng of the GNN classifier on augmented graphs. The second stage of classification follows general GNN training setup, without introducing additional complexity: the time/memory complexity per layer of using Graph Isomorphism Networks (GIN) as backbone is $\Theta(n+e)$, where $n$ is the number of nodes and $e$ is the number of edges. For the data augmentation stage, we introduce graph transformer whose memory and time complexity per layer is $\Theta(n^2)$. This arises from the computation of attention scores and predictions across each edge.

