# OpenReview forum: "Mitigating Graph Covariate Shift via Score-based Out-of-distribution Augmentation"
_ICLR.cc/2025/Conference — Submitted to ICLR 2025_

### Official Review · Reviewer_HMbi · 2024-10-18

**Soundness:** 2
**Presentation:** 3
**Contribution:** 2
**Rating:** 6
**Confidence:** 4

**Summary:**

This paper introduces a innovative OOD Augmentation (OODA) method to address the issue of covariate shifts in graph learning. Specifically, the authors propose a conditional score-based graph generation strategy that synthesizes unseen graphs. Extensive experiments show that OODA outperforms many baselines on synthetic and real-world datasets, including molecular and sentiment analysis tasks, under various covariate shift conditions.

**Strengths:**

1. The paper is easy to follow with clear logic flow.
2. Using diffusion models to generate OOD examples for OOD generalization is interesting.
3. The experiments conducted are comprehensive with non-trivial improvements.

**Weaknesses:**

Although I find the diffusion-based OOD augmentation intriguing, there are several major weaknesses.

1. The understanding of the OOD literature is not deep or accurate enough, which leads to misleading descriptions. For example, [1] is misused in the statement “line 162: separating environmental components could be inherently unfeasible,” while the original paper actually indicates that without explicit environmental information and **extra assumptions**, it is infeasible to achieve OOD generalization.
2. While previous methods generally introduced clear assumptions and theoretical guarantees, this paper does not include any theoretical analysis. As a result, it is unclear why the proposed method would effectively solve the OOD generalization problem.
3. The authors claim that covariate shifts are relatively neglected in graph learning, but they missed an important covariate shift-related baseline [2]. I believe including a comparison with this method would make the performance results more convincing. To be clear, I don’t expect this method to outperform those that leverage environment labels. However, it is crucial to compare the performance under your assumptions and those of environment-available baselines. The performance of a good method under OOD assumptions can approach that of environment-available methods. *Essentially, with proper theoretical guarantees for optimization, comparing different OOD methods becomes a competition of assumptions.*

[1] Chen, Yongqiang, Yatao Bian, Kaiwen Zhou, Binghui Xie, Bo Han, and James Cheng. “Does invariant graph learning via environment augmentation learn invariance?” Advances in Neural Information Processing Systems 36 (2023).

[2] Gui, Shurui, Meng Liu, Xiner Li, Youzhi Luo, and Shuiwang Ji. “Joint learning of label and environment causal independence for graph out-of-distribution generalization.” Advances in Neural Information Processing Systems 36 (2023).

**Questions:**

1. Will this generation process eliminate important components of graphs?
2. Is the OOD guidance targeting generating samples with low likelihood under the original distribution? Is it the only assumption used? Any other constraints?
3. What do you mean more uniform relative to the original data distribution? (line 279)

---

> ### Author Response · Authors · 2024-11-23
> **Reponse to Reviewer HMbi -- Part 1**
>
> Thank you for your insightful comments and requests for clarifications. We will try addressing your concerns below:
>
> >**W1**: The understanding of the OOD literature is not deep or accurate enough, which leads to misleading descriptions.
>
>
> Thank you for pointing out the discrepancy in our description. We appreciate your guidance on ensuring our references accurately reflect the literature.
>
> We will revise the statement to accurately reflect the insights from [1]. The new sentence will read: "Without explicit environmental information and additional assumptions, separating environmental components could be inherently unfeasible." This aligns with the authors in [1], who state: "Invariant and spurious subgraphs are denoted as $G_c$ and $G_s$. More formally, if $\exists G_s$, such that $P^{e_1}\left(Y \mid G_s\right)=P^{e_2}\left(Y \mid G_s\right)$ for any $e_1, e_2 \in \mathcal{E}_{\text {tr }}$,
>
> where $P^e\left(Y \mid G_s\right)$ is the conditional distribution $P\left(Y \mid G_s\right)$ under environment $e \in \mathcal{E}_{\text {all }}$, it is impossible for any graph learning algorithm to identify $G_c$."

---

> ### Author Response · Authors · 2024-11-23
> **Reponse to Reviewer HMbi -- Part 2**
>
> >**W2**: This paper does not include any theoretical analysis. As a result, it is unclear why the proposed method would effectively solve the OOD generalization problem.
>
> We agree on the importance of theoretical gaurantee and acknowledge the challenge of providing one given the complexity of the SDE procedure. We would like to highlight the pioneering contribution of our work in designing a **tractable conditional reverse-time SDE** framework to mitigate graph covariate shifts, offering a fresh perspective on addressing this significant challenge in the field. The design of our SDE is carefully justified with clear principles, and the empirical results demonstrates a clear improvement. We believe the empricial and practical contribution of this work should not be diminished. We want to further explain the principles that support our design.
>
> **Prior Knowledge of Out-of-Distribution (OOD) Samples**: As outlined in [5], our prior knowledge about the out-of-distribution aspect of input graphs $G$ is that they comprise two distinct components: the causal variable $C$ and the non-causal variable $S$, such as the House motif and the Tree base in the GOOD-Motif dataset. Here, $C$ serves as the sole endogenous parent crucial for determining the ground-truth label $y$, while $S$ is an element that varies across different environments. When synthesizing OOD samples, we use the label variable $y$ as prior knowledge and conditions to control the diffusion procudure, detailed as follows.
>
>
> **Conditional Reverse-time SDE with Label Knowledge**: Our approach specifically maintains the integrity of the causal part $C$ while exploring the unstable non-causal part $S$ through the application of a conditional reverse-time Stochastic Differential Equation (SDE), as presented in Eq. (6). In Eq. (6), the condition $y_G$ controls the generated graph to maintain the label knowledge (i.e., keeping causal part $C$), and the condition $y_\text{ood}$ augments the graph to explore the environment (i.e., varying non-causal part $S$). This equation has never been proposed before, and the solution balancing label preservation and enrivonment exploration is nontrivial. To model this, we propose formulating the distribution $p_t\left(\mathbf{y}_{\text {ood }}=\lambda \mid G_t, \mathbf{y}_G\right)$ as proportional to the negative exponent of the joint density of $G_t$ and $\mathbf{y}_G, p_t\left(G_t, \mathbf{y}_G\right)$. This derives the expression for the gradient of the log probability for conditional reverse diffusion, as presented in Eq. (9). Eq. (9) is featured with the following properties:
>
> **Preserving the Label Knowledge by $y_G$**: The second term $\log p_t\left(\mathbf{y}_G \mid G_t\right)$ aims to maximize the prediction probability of the graph's ground-truth label $y_G$ for each step. This essentially maintians the prior label knowledge when synthesizing the OOD samples.
>
> **Exploring the Training Distribution by $y_\text{ood}$**: Combining two terms, the conditional score function creates a marginal distribution proportional to $p_t\left(G_t, \mathbf{y}_G\right)^{1-\sqrt{\lambda}}$.
>
> When environmental graphs are explored, their patterns exist in the low-density regions of the original distribution. Conversely, un-explored graphs exist in high-density regions. With exploration strength $\lambda=0$, the marginal distribution becomes $p_t\left(G_t, \mathbf{y}_G\right)$,
>
> which denoises the synthesized graphs from the augmented distribution to follow the original distribution $P_{\operatorname{tr}}(G, Y)$, thus both casual part $C$ and non-casaul part are preserved. By adjusting $\lambda$, we regulate dispersion, allowing generated graphs to maintain the causal $C$ and explore non-causal $S$, thus increasing the divergence between $P_{\mathrm{tr}}\left(G, Y \mid \mathbf{y}_{\text {ood }}=\lambda\right)$
>
> and $P_{\mathrm{tr}}(G, Y)$.
>
>
> Thus, our formulation achieves a **tractable conditional reverse-time SDE**. This advancement not only simplifies the generating OOD graphs by converting it into a conditional generation problem, but also presents a novel application of reverse-time SDE techniques in this context of graph covariate shifts.

---

> ### Author Response · Authors · 2024-11-23
> **Reponse to Reviewer HMbi -- Part 3**
>
> >**W3**: An important covariate shift-related baseline [4] should be compared, though this method leverages environment labels.
>
> We acknowledge the necessity of including a performance comparison with the baseline [4] related to covariate shifts that you mentioned. In the final version of our paper, we will incorporate this baseline and thoroughly discuss the comparative results. This will allow us to provide a comprehensive view of how our method performs under our specific assumptions versus those that leverage additional environment labels. Below is the comparison.
>
>
> | Models | GOOD-Motif-base | GOOD-Motif-size | GOOD-CMNIST-color | GOOD-HIV-scaffold | GOOD-HIV-size | GOOD-SST2-length |
> | -------- | -------- | -------- | -------- | -------- | -------- | -------- |
> |   LECI   | $84.56 \pm 2.22$     | $71.43 \pm 1.96$      | $51.80 \pm 2.70$     | $74.43 \pm 1.69$      | $65.44 \pm 1.78$      | $83.44 \pm 0.27$      |
> |   **OODA (ours)**   | $75.25 \pm 3.84$     | $60.81 \pm 7.80$     | $54.60 \pm 2.27$     | $72.67 \pm 1.28$      | $66.47 \pm 2.29$      | $82.69 \pm 0.28$      |
>
> As shown in the table, OODA achieves performance comparable to that of environment-aware methods on the GOOD-CMNIST, GOOD-HIV, and GOOD-SST2 datasets. We believe it would be a valuable future work to further improve the OOD performance by incorporating environment labels into our framework.
>
> >**Q1**: Will this generation process eliminate important components of graphs?
>
> By design, we aim to preserve important graph components (i.e., the label information) during the generation process via the modeling of $\log p_t\left(\mathbf{y}_G \mid G_t\right)$, as the second term in Eq. (9) shows.
>
> Our evaluations, presented in Figures 3, 4, and 5, verify OODA's capability to generate OOD graphs while effectively retaining stable patterns. This is evidenced by the expected probability $\mathbb{E}_{\tilde{G}_0 \sim \tilde{P}}\left[p\left(\mathbf{y}_G \mid \tilde{G}_0\right)\right]$ consistently exceeds $0.90$ as $\lambda$ increases. Additionally, visualizations in Table 3 demonstrate that as $\lambda$ increases, there are gradual modifications to the base graph structures, while key motifs remain intact. Finally, results in Table 1 illustrate that OODA outperforms state-of-the-art methods across diverse datasets. Such improvements would not be achievable if important graph components were eliminated during the generation process. The retention of these components is crucial for enhancing model performance and ensuring the reliability of generated graphs.
>
> >**Q2**: Is the OOD guidance targeting generating samples with low likelihood under the original distribution? Is it the only assumption used? Any other constraints?
>
> Yes, the OOD guidance indeed aims at generating samples with low likelihood under the original distribution.
>
> In addition to focusing on low-likelihood samples, we adopt the assumption, based on [2], that input graphs $G$ consist of two distinct components: the causal part $C$ and the non-causal part $S$. For instance, in datasets like GOOD-Motif, the House motif and Tree base exemplify this structure. The causal part $C$ is considered the sole endogenous parent that determines the ground-truth label $Y$, while the non-causal part $S$ varies across different environments, represented as  $C \rightarrow Y$. Therefore, OODA uses a novel guidance scheme that generates augmented graphs, concurrently retaining predictive stable patterns and incorporating explored environments.
>
> As highlighted in [3], ensuring meaningful exploration requires keeping deviations within a certain range from the original distribution to maintain the integrity of the causal component $C$. Therefore, the choice of $\lambda$ is crucial and must be meticulously calibrated based on performance metrics from OOD validation sets across diverse datasets to ensure effective and responsible exploration.

---

> ### Author Response · Authors · 2024-11-23
> **Reponse to Reviewer HMbi -- Part 4**
>
> >**Q3**: What do you mean more uniform relative to the original data distribution? (line 279)
>
> By "more uniform relative to the original data distribution," we refer to a broadening or dispersion of the original distribution $P_{\mathrm{tr}}\left(G, Y\right)$. We have changed the original statement in the modified version to avoid confusion. In the original distribution, the data for a class $Y$ often appears as a tightly centered cluster within the data space, with well-defined causal relationships. When we increase $\lambda$, it modulates this distribution to enhance variability and exploration. As $\lambda$ increases, it influences the distribution to be more dispersed. This enhanced dispersion allows for the sampling of graphs that retain the critical causal components responsible for determining label $Y$. However, these graphs also incorporate more diverse and novel non-causal elements, sampled from $P_{\mathrm{tr}}\left(G, Y \mid \mathbf{y}_{\mathrm{ood}}=\lambda\right)$. This enables the inclusion of a wider array of graph features, fostering exploration beyond the tightly clustered native structures, which can lead to novel insights and robust model learning.
>
> **References:**
>
> [1] Chen, Yongqiang, Yatao Bian, Kaiwen Zhou, Binghui Xie, Bo Han, and James Cheng. “Does invariant graph learning via environment augmentation learn invariance?” Advances in Neural Information Processing Systems 36 (2023).
>
> [2] Wu, et al. Discovering invariant rationales for graph neural networks. https://arxiv.org/abs/2201.12872
>
> [3] Sui, et al. Unleashing the Power of Graph Data Augmentation on Covariate Distribution Shift. https://arxiv.org/abs/2211.02843
>
> [4] Gui, Shurui, Meng Liu, Xiner Li, Youzhi Luo, and Shuiwang Ji. “Joint learning of label and environment causal independence for graph out-of-distribution generalization.” Advances in Neural Information Processing Systems 36 (2023).

---

> > ### Comment · Reviewer_HMbi · 2024-11-25
> >
> > Thanks for the authors rebuttals. Parts of my major concerns have been addressed, and I have raised my score from 5 to 6. However,  it is worth noting that I believe the theoretical analysis can be formulated better and more formal in the manuscript.

---

### Official Review · Reviewer_CufZ · 2024-10-27

**Soundness:** 2
**Presentation:** 3
**Contribution:** 2
**Rating:** 5
**Confidence:** 4

**Summary:**

This study proposes a novel framework called Out-of-Distribution Diffusion Augmentation (OODA), which employs score-based graph generation to address covariate distribution shifts in graph data. The core idea is to expand the training distribution by generating new graph samples that capture previously unseen environmental features while preserving stable, predictive patterns, thus enhancing the model’s OOD generalization capabilities.

**Strengths:**

1. The paper is well-organized, clearly motivating the need for improved graph OOD generalization by highlighting the limitations of existing perturbation-based augmentations.

2. Extending the score-based diffusion model to graph OOD generalization is novel, which aims to enable controlled augmentation that balances diversity with stability.

3. Extensive experiments demonstrate the effectiveness of the proposed OODA framework. The experiments highlight the practical relevance of the method for real-world applications involving covariate shifts in graph data.

**Weaknesses:**

1. A primary concern is the diversity of the generated graphs, which remains somewhat unclear. While the authors report that the MMD distance increases as $\lambda$ grows (Sec. 5.3), this metric alone does not sufficiently convey how diverse the generated subgraphs are for handling graph covariate shifts. The authors should conduct experiments to demonstrate the quality of the generated samples quantitatively (using some metric to quantify the degree of diversity), and better to provide a theoretical analysis as this is the key component of this work.

2. The framework’s effectiveness in addressing size shifts is unclear, as the generated subgraphs are constrained to match the original subgraphs in the total node count if I understand correctly. Despite this, the model performs well on the GOOD-HIV-size benchmark, can the authors explain how and why the method can generalize well for size shift on GOOD-HIV-size dataset but not so well on GOODMotif-size dataset? Second, OODA perform similarly to the SOTA method such AIA on synthetic datasets, e.g., GOOD-Motif. This may indicate that the samples generated by this method have limitations in terms of accuracy and diversity.

3. Experimental comparisons may not be fair, as prior methods use GNN as backbones, whereas OODA relies on multiple graph transformers for estimating score functions and for predicting class labels. Furthermore, training multiple graph trasnsformers also introduce additional time and space complexity. Can the authors discuss the time and space complexity, and report the time and memory consumption, and compare with other methods?

4. The choice to use a pretrained graph transformer for analyzing stable pattern preservation (line 435) lacks clarity, why not use the finally obtained invariant GNN for evaluation, but use the graph trasnformer,which is utilized for data augmentation.

5. $\lambda$ would affect the generated graphs’ smoothness and the degree of diversity, yet the paper lacks a sensitivity analysis for this parameter. How different values of $\lambda$ affect the graph diversity and model performance?

6. OODA modify the differential equation to incorporate labels in Eqn. 5, however, it lacks a theoretical guarantee for the generated samples to preserve stable patterns and promote the diversity. It is better to provide some theoretical insights on this aspect.

**Questions:**

1. $\lambda$ controls the extent of the exploration and the diversity of the generated sample, does it mean higher $\lambda$ will lead to better OOD generalization performance?

---

> ### Author Response · Authors · 2024-11-23
> **Reponse to Reviewer CufZ -- Part 1**
>
> Thank you for your insightful comments and requests for clarifications. We now try to address your concern below:
>
> >  **W1**: Quantify the diversity of the generated graphs.
>
> Thank you for highlighting the importance of assessing the diversity of the generated graphs. We appreciate the opportunity to clarify our approach and provide additional details to address this key component of our work.
>
> **MMD as a quantitative diversity metric**: In [1], comprehensive experiments were conducted to rigorously evaluate and benchmark metrics for assessing the diversity of generated graphs. In its Section 4.2, it highlights the correlation between random Graph Isomorphism Networks (GIN)-based Maximum Mean Discrepancy (MMD) with RBF kernel and the diversity of generated graphs, demonstrating the rationale of why MMD can serve as a reasonable diversity metric. Consequently, we employ this metric to assess the discrepancy between augmented and training distributions. As illustrated in Figures 2 and 5, MMD with the RBF kernel increases as $\lambda$ grows, indicating an enhanced diversity as the exploration strength increases. Additionally, we provide examples of the generated OOD graphs in Table 3, offering further qualitative evidence of their diversity. Similarly, Figure 8 in [2] employs visualization to further evaluate and substantiate the diversity of the generated graphs.
>
> **FCD as a diversity metric for molecules**: As demonstrated in [3], the Fréchet ChemNet Distance (FCD) metric incorporates chemically and biologically relevant information about molecules and assesses the diversity of generated sets through their distribution. Consequently, we utilize FCD to evaluate the discrepancy between augmented and training molecules. As illustrated in Figure 4, FCD increases with higher values of $\lambda$, also matching the previous conclusion under MMD metric.
>
> In addition to these two metrics, we are more than happy to include more if the reviewer have suggestions on any specific diversity metrics.
>
> >**W2**: The framework’s effectiveness in addressing size shifts is unclear.
>
> As detailed in Appendix A.4.2, we have employed a flexible approach to account for size variations across different datasets. Specifically, for GOOD-SST2-length and GOOD-Motif-size, where $\lambda = 0.01$ corresponds to an increase of one node, we employed a grid search with $\lambda \in\{0.01,0.02, \ldots, 0.14\}$ and $\lambda \in\{0.01,0.02,0.03,0.04,0.05\}$, respectively. For GOOD-HIV-size, $\lambda = 0.01$ represents a decrease of ten nodes, utilizing the same $\lambda$ values as GOOD-Motif-size. This method allows generated subgraphs to vary in node count, rather than being constrained to match the original subgraph sizes.
>
> Our model demonstrated a $10.89\%$ performance improvement on the GOOD-HIV-size benchmark over empirical risk minimization (ERM). Additionally, it achieved a $17.53\%$ performance improvement on the GOOD-Motif-size dataset. Therefore, the model performs effectively across both the GOOD-HIV-size and GOODMotif-size datasets. This indicates its robust generalization capacity in handling size shifts effectively.
>
> OODA is expressly crafted to tackle not only shifts in graph structures but also variations in node and edge features. This multi-faceted capability allows OODA to effectively handle covariate shifts across both feature and structural distributions simultaneously. In comparison, its performance is analogous to the state-of-the-art method AIA [4] (NeurIPS 2023) on synthetic datasets like GOOD-Motif. While AIA improved upon the SOTA method VREx by $3.04\%$ on the GOOD-Motif-base dataset, OODA outperformed AIA by $2.19\%$. This reflects a competitive enhancement, demonstrating that both methods achieve comparable progress, yet with differing strengths in specific contexts.

---

> ### Author Response · Authors · 2024-11-23
> **Reponse to Reviewer CufZ -- Part 2**
>
> >**W3**: Experimental comparisons may not be fair, as prior works use GNN as backbone and OODA uses graph transformer. What are the time and space complexity?
>
> **Fair comparison using GNN backbone**: Thank you for bringing attention to clarify the experimental setups. We would like to highlight that in our work, we trained graph transformers only to compute the conditional reverse-time SDE, and after using the trained diffusion model to generate augmented graphs, **we still follow prior works to use the same GNN architecture as backbones for the graph classification task**, as elaborated in Appendix A.4.2.
>
> **Time and memory complexity**: Our pipeline consists of two stages: data augmentation, and trainng of the GNN classifier on augmented graphs. The second stage of classification follows general GNN training setup, without introducing additional complexity: the time/memory complexity per layer of using Graph Isomorphism Networks (GIN) as backbone is $\Theta(n + e)$, where $n$ is the number of nodes and $e$ is the number of edges. For the data augmentation stage, we introduce graph transformer whose memory and time complexity per layer is $\Theta(n^2)$. This arises from the computation of attention scores and predictions across each edge.
>
> >**W4**: The choice to use a pretrained graph transformer for analyzing stable pattern preservation (line 435) lacks clarity, why not use the finally obtained invariant GNN for evaluation, but use the graph trasnformer, which is utilized for data augmentation.
>
> We apologize for the confusion. The goal of this experiment (Figure 3), along with the experiment for Figure 2, is to demonstrate whether our graph generator achieves the two desired properties: preserving stable patterns (i.e., the generated graph preserves its ground-truth label), and exploring environment patterns (i.e., the generated distribution is distant from training distribution). Since the graph transformer is to model $p_t\left(\mathbf{y}_G \mid G_t\right)$ in the augmentation stage, it is more reasonable to directly use its accuracy to verify the label information is effectively captured in the reverse-time SDE procedure.
>
> >**W5**: How different values of $\lambda$ affect the graph diversity and model performance?
>
> We have conducted a comprehensive visualization of the generated OOD graphs under various $\lambda$ values, as depicted in Table 3. This table illustrates that increasing $\lambda$ systematically alters the structures of the base graphs, while the core motifs remain consistently preserved. This indicates that $\lambda$ plays a crucial role in modulating structural modifications without compromising key patterns.
>
> The choice of $\lambda$ is informed by its evaluated performance on OOD validation sets across a range of datasets, as detailed in Appendix A.4.2. Below are additional ablation results to showcase the impact of different $\lambda$ values on model performance:
>
>
> | $\lambda$ | GOOD-Motif-base | GOOD-HIV-scaffold |
> | -------- | -------- | -------- |
> |  0.01   |   91.80   | 78.45   |
> |  0.02   | 92.97     | 78.50     |
> |  0.03   | 92.83     | 79.49     |
> |  0.04   | 93.03     | 78.71     |
> |  0.05   | 92.77    | 78.69   |
>
>
> | $\lambda$ | GOOD-CMNIST-color |
> | -------- | -------- |
> |  0.05   |   68.66   |
> |  0.1   | 67.91     |

---

> ### Author Response · Authors · 2024-11-23
> **Reponse to Reviewer CufZ -- Part 3**
>
> >**W6**: It is better to provide some theoretical insights for the generated samples to preserve stable patterns and promote the diversity.
>
> We agree on the importance of theoretical gaurantee and acknowledge the challenge of providing one given the complexity of the SDE procedure. We would like to highlight the pioneering contribution of our work in designing a **tractable conditional reverse-time SDE** framework to mitigate graph covariate shifts, offering a fresh perspective on addressing this significant challenge in the field. The design of our SDE is carefully justified with clear principles, and the empirical results demonstrates a clear improvement. We believe the empricial and practical contribution of this work should not be diminished. We want to further explain the principles that support our design.
>
> **Prior Knowledge of Out-of-Distribution (OOD) Samples**: As outlined in [5], our prior knowledge about the out-of-distribution aspect of input graphs $G$ is that they comprise two distinct components: the causal variable $C$ and the non-causal variable $S$, such as the House motif and the Tree base in the GOOD-Motif dataset. Here, $C$ serves as the sole endogenous parent crucial for determining the ground-truth label $y$, while $S$ is an element that varies across different environments. When synthesizing OOD samples, we use the label variable $y$ as prior knowledge and conditions to control the diffusion procudure, detailed as follows.
>
>
> **Conditional Reverse-time SDE with Label Knowledge**: Our approach specifically maintains the integrity of the causal part $C$ while exploring the unstable non-causal part $S$ through the application of a conditional reverse-time Stochastic Differential Equation (SDE), as presented in Eq. (6). In Eq. (6), the condition $y_G$ controls the generated graph to maintain the label knowledge (i.e., keeping causal part $C$), and the condition $y_\text{ood}$ augments the graph to explore the environment (i.e., varying non-causal part $S$). This equation has never been proposed before, and the solution balancing label preservation and enrivonment exploration is nontrivial. To model this, we propose formulating the distribution $p_t\left(\mathbf{y}_{\text {ood }}=\lambda \mid G_t, \mathbf{y}_G\right)$ as proportional to the negative exponent of the joint density of $G_t$ and $\mathbf{y}_G, p_t\left(G_t, \mathbf{y}_G\right)$. This derives the expression for the gradient of the log probability for conditional reverse diffusion, as presented in Eq. (9). Eq. (9) is featured with the following properties:
>
> **Preserving the Label Knowledge by $y_G$**: The second term $\log p_t\left(\mathbf{y}_G \mid G_t\right)$ aims to maximize the prediction probability of the graph's ground-truth label $y_G$ for each step. This essentially maintians the prior label knowledge when synthesizing the OOD samples.
>
> **Exploring the Training Distribution by $y_\text{ood}$**: Combining two terms, the conditional score function creates a marginal distribution proportional to $p_t\left(G_t, \mathbf{y}_G\right)^{1-\sqrt{\lambda}}$.
>
> When environmental graphs are explored, their patterns exist in the low-density regions of the original distribution. Conversely, un-explored graphs exist in high-density regions. With exploration strength $\lambda=0$, the marginal distribution becomes $p_t\left(G_t, \mathbf{y}_G\right)$,
>
> which denoises the synthesized graphs from the augmented distribution to follow the original distribution $P_{\operatorname{tr}}(G, Y)$, thus both casual part $C$ and non-casaul part are preserved. By adjusting $\lambda$, we regulate dispersion, allowing generated graphs to maintain the causal $C$ and explore non-causal $S$, thus increasing the divergence between $P_{\mathrm{tr}}\left(G, Y \mid \mathbf{y}_{\text {ood }}=\lambda\right)$
>
> and $P_{\mathrm{tr}}(G, Y)$.
>
>
> Thus, our formulation achieves a **tractable conditional reverse-time SDE**. This advancement not only simplifies the generating OOD graphs by converting it into a conditional generation problem, but also presents a novel application of reverse-time SDE techniques in this context of graph covariate shifts.
>
> **References:**
>
> [1] Thompson, et al. On Evaluation Metrics for Graph Generative Models. https://arxiv.org/abs/2201.09871
>
> [2] Chen, et al. D4Explainer: In-Distribution GNN Explanations via
> Discrete Denoising Diffusion. https://arxiv.org/abs/2310.19321
>
> [3] Preuer, et al. Fréchet ChemNet Distance: A metric for generative models for molecules in drug discovery. https://arxiv.org/abs/1803.09518
>
> [4] Sui, et al. Unleashing the Power of Graph Data Augmentation on Covariate Distribution Shift. https://arxiv.org/abs/2211.02843
>
> [5] Wu, et al. Discovering invariant rationales for graph neural networks. https://arxiv.org/abs/2201.12872

---

> > ### Comment · Reviewer_CufZ · 2024-11-25
> >
> > Thank you for your detailed reply and new experimental results. However, my concerns remain as follows:
> >
> > - **Time complexity:** This method requires multiple graph transformers, which significantly increases training time compared to previous methods. This hinders the method's applicability to larger datasets.
> >
> > - **Use of transformers:** While the final model training employs the same GNN architecture, utilizing graph transformers for data augmentation introduces some unfairness, as other methods like GREA and AIA rely on GNNs for data augmentation.
> >
> > - **Effectiveness under strong spurious correlations:** Although $\log p_t\left(\mathbf{y}_G \mid G_t\right)$ incorporates labels to preserve the invariant subgraph, when spurious correlations are strong, it is likely that spurious subgraphs will also be preserved. This limits the method's effectiveness.
> >
> > - **Impact of $\lambda$ on diversity:** The parameter $\lambda$ controls the diversity of the generated graphs. However, when $\lambda$ increases, the test performance does not improve, which is inconsistent with the method's motivation.
> >
> > For these reasons, I have to maintain my score.

---

### Official Review · Reviewer_FKFr · 2024-11-03

**Soundness:** 2
**Presentation:** 3
**Contribution:** 2
**Rating:** 5
**Confidence:** 4

**Summary:**

This work studies the data augmentation method for graph OOD generalization under covariate shifts. The authors argue that previous approaches in graph augmentation may generate inaccurate samples, as which heavily rely on an accurate separation of stable and environmental features. Therefore, they propose a new approach that adopts score-based graph generation strategies to synthesize unseen environmental features while preserving the validity and stable features of the overall graph patterns. They conduct extensive experiments on GOOD benchmark to verify the effectiveness of the proposed approach.

**Strengths:**

(+) This work identifies a critical problem in generating valid graph examples for graph OOD generalization.

(+) The idea of adopting a score-based strategy is new and interesting.

(+) The experiments show the effectiveness of the proposed method.

**Weaknesses:**

(-) It remains unclear to what extent the proposed score-based strategy could preserve the stable features.

(-) The technical novelty is limited.

(-) The ablation studies are limited.

**Questions:**

1. It remains unclear to what extent the proposed score-based strategy could preserve the stable features. It seems that the preservation of the stable graph features relies on the classifier $\phi_t$.
- However, it is unclear how the classifier is trained;
- If there exists a classifier that can already reliably approximate $p_t(y_G|G_t)$, then, it seems the classifier itself can already be used for tackling covariate shifts;
- In addition, it has been mentioned several times about the variable $y_ood$, but it has not been formally defined. Meanwhile, it is unclear why $y_ood$ could specify the amount of OOD exploration;

2. The technical novelty is limited, as it is a simple modification of the diffusion models on graphs, and no additional theoretical results are provided;

3. The ablation studies are limited:
- It is unclear how the hyperparameters are selected and whether OODA is sensitive to the choice of hyperparameters;
- It'd be great to give more examples of generated realistic graphs;

4. Can we use the generated graphs to improve the performance of other graph OOD methods?

---

> ### Author Response · Authors · 2024-11-23
> **Reponse to Reviewer FKFr -- Part 1**
>
> Thank you for your constructive comments and requests for clarifications. We respond to your concerns below:
>
> > **W1 & Q1:** To what extend the proposed strategy could preserve the stable features.
>
> We appreciate the detailed questions to clarify the design of our method.
>
> **How the classifier is trained:** First, the classifier is trained using **noisy graphs** (i.e., $G_t$) sampled at random time steps to model $p_t\left(\mathbf{y}_G \mid G_t\right)$. During this process, we split the OOD training data into separate training and validation sets, selecting the classifier that exhibits the best performance on the validation set. Note that **neither the OOD validation set nor the OOD test set** is utilized during this training phase. Figure 3 demonstrates the predictive probability of the classifier on the ground-truth label, and a consistant high value (>0.95) highlights the ability of the proposed strategy in preserving the stable label information.
>
> **Can the classifier itself already be used for tackling covariate shifts:** Although the classifier is instrumental in modeling $p_t\left(\mathbf{y}_G \mid G_t\right)$, it is not employed to directly address covariate shifts of $p_0\left(\mathbf{y}_G \mid G_0\right)$, where $G_0$ represents the clean graph. The classifier's training on noisy graphs aims to capture the underlying feature dynamics under distorted conditions, without presupposing reliability in tackling covariate shift scenarios independently. To further illustrate that the classifier alone cannot effectively tackle covariate shifts, we present its performance on OOD test data. The results indicate that the classifier's performance is lower than that achieved by empirical risk minimization (ERM), confirming its limitations in addressing covariate shifts as it is not designed to do so.
>
> Models | GOOD-Motif-base | GOOD-SST2-length |
> | -------- | -------- | -------- |
> |  ERM  | $68.66$     |  $81.30$     |
> |  Only the classifier $\phi_t$  | $62.63$     |  $80.54$     |
> |  OODA  | $75.25$     | $82.69$     |
>
> **Why $y_\text{ood}$ could specify the amount of OOD exploration:** Our approach utilizes the understanding that OOD samples are characterized by low likelihood, as demonstrated in [1, 2, 3, 4]. Following this principle, we model the distribution $p_t\left(\mathbf{y}_{\text {ood }}=\lambda \mid G_t, \mathbf{y}_G\right)$ to be inversely proportional to the joint density of $G_t$ and $\mathbf{y}_G$, denoted as:
>
> $$p_t\left(\mathbf{y}_{\text {ood }}=\lambda \mid G_t, \mathbf{y}_G\right) \propto p_t\left(G_t, \mathbf{y}_G\right)^{-\sqrt{\lambda}}=p_t\left(G_t\right)^{-\sqrt{\lambda}} p_t\left(\mathbf{y}_G \mid G_t\right)^{-\sqrt{\lambda}}.$$
>
> Intuitively, as $\lambda$ approaches 1, this distribution sharpens, with the negative exponent accentuating conditions indicative of OOD instances. This characteristic allows $\lambda$ to calibrate the extent of OOD exploration by amplifying the emphasis on regions in the distribution where the probability values are inherently smaller, effectively modeling the OOD phenomena.
>
> We hope these clarifications address your concerns, and we are open to further discussion and questions regarding our approach.

---

> ### Author Response · Authors · 2024-11-23
> **Reponse to Reviewer FKFr -- Part 2**
>
> > **W2 & Q2**: The technical novelty is limited, as it is a simple modification of the diffusion models on graphs, and no additional theoretical results are provided.
>
> We respectfully argue that our approach offers more than a simple modification of diffusion models on graphs. As demonstrated in Section 4, we derive a novel and desired conditional reverse-time Stochastic Differential Equation (SDE), as presented in Eq. (6). Eq. (6) has never been proposed before, and the solution balancing label preservation and enrivonment exploration is nontrivial. To model this, we propose formulating the distribution $p_t\left(\mathbf{y}_{\text {ood }}=\lambda \mid G_t, \mathbf{y}_G\right)$ as proportional to the negative exponent of the joint density of $G_t$ and $\mathbf{y}_G, p_t\left(G_t, \mathbf{y}_G\right)$. This derives the expression for the gradient of the log probability for conditional reverse diffusion, as presented in Eq. (9). In Eq. (9), when $\lambda$ increases from 0, the augmented distribution becomes broader compared to the original data distribution $P\left(G \mid \mathbf{y}_G\right)$. Consequently, the generated samples are more likely to originate from the out-of-distribution context of $P\left(G \mid \mathbf{y}_G\right)$.
>
> Thus, our formulation achieves a **tractable conditional reverse-time SDE**. This advancement not only simplifies the generating OOD graphs by converting it into a conditional generation problem, but also presents a novel application of reverse-time SDE techniques in this context of graph covariate shifts.

---

> ### Author Response · Authors · 2024-11-23
> **Reponse to Reviewer FKFr -- Part 3**
>
> > **W3 & Q3**: The ablation studies are limited.
>
> Thank you for your insightful question regarding the hyperparameter and generated graph examples in the ablation studies.
>
> **Sensitivity of hyperparameter $\lambda$**: We understand the importance of hyperparameter selection and its impact on model performance. As detailed in Appendix A.4.2, we determine the value of $\lambda$ by evaluating its effectiveness on OOD validation sets across various datasets. Below are the ablation results that illustrate the sensitivity of our method to different $\lambda$ values:
>
>
> | $\lambda$ | GOOD-Motif-base | GOOD-HIV-scaffold |
> | -------- | -------- | -------- |
> |  0.01   |   91.80   | 78.45   |
> |  0.02   | 92.97     | 78.50     |
> |  0.03   | 92.83     | 79.49     |
> |  0.04   | 93.03     | 78.71     |
> |  0.05   | 92.77    | 78.69   |
>
>
> | $\lambda$ | GOOD-CMNIST-color |
> | -------- | -------- |
> |  0.05   |   68.66   |
> |  0.1   | 67.91     |
>
> **More realistic graph examples**: The reason we demonstrate generated graphs for the synthetic data GOOD-Motif-base is that the ground-truth casual part is known and intuitive to verify. In addition to the GOOD-Motif-base graphs, we are currently compiling and visualizing a diverse set of graph examples for other datasets in an intuitive way, and will include them in the final version of the paper once available.
>
> >**Q4**: Can we use the generated graphs to improve the performance of other graph OOD methods?
>
> Thank you for your insightful question about the potential of our OOD generation method. We have explored this application by incorporating our generated graphs with one of the state-of-the-art method, AIA [7]: applying AIA upon the augmented graph set produced by our method. Our preliminary results indicate that integrating our generated graphs with existing methodologies can indeed lead to measurable performance improvements.
>
>
> | Models | GOOD-SST2 | GOOD-CMNIST | GOOD-Motif-base |
> | -------- | -------- | -------- | -------- |
> | AIA   | $81.69 \pm 0.57$     | $36.37 \pm 4.44$     | $73.64 \pm 5.15$     |
> | **AIA with our OODA**  | $83.10 \pm 0.46$    | $39.76 \pm 5.58$     | $74.03 \pm 5.49$     |
>
> These initial findings demonstrate that our approach has the potential to complement and enhance the effectiveness of other established graph OOD methods like AIA. By leveraging the diversity and novelty of the graphs produced by our model, we can offer an enriched training data set that strengthens the robustness and generalization capabilities of these methods.
>
> **References**:
>
> [1] Du, et al. Implicit Generation and Modeling with Energy-Based
> Models. https://arxiv.org/abs/1903.08689
>
> [2] Grathwohl, et al. Your Classifier is Secretly an Energy Based Model and You Should Treat it Like One. https://arxiv.org/abs/1912.03263
>
> [3] Wu, et al. Energy-based Out-of-Distribution Detection for Graph Neural Networks. https://arxiv.org/abs/2302.02914
>
> [4] Bazhenov, et al. Towards OOD Detection in Graph Classification from Uncertainty Estimation Perspective. https://arxiv.org/abs/2206.10691
>
> [5] Wu, et al. Discovering invariant rationales for graph neural networks. https://arxiv.org/abs/2201.12872
>
> [6] Liu, et al. Data-Centric Learning from Unlabeled Graphs with Diffusion Model. https://arxiv.org/abs/2303.10108
>
> [7] Sui, et al. Unleashing the Power of Graph Data Augmentation on Covariate Distribution Shift. https://arxiv.org/abs/2211.02843

---

> > ### Comment · Reviewer_FKFr · 2024-11-26
> >
> > Thank you for reading and addressing my concerns in the review carefully. Nevertheless, it seems the following points remain not clearly explained:
> > - Could you elaborate more on why the classifier could capture the semantic information?
> > - As the classifier is used for both identifying the semantic information and instructing the diffusion process, would not the whole training turn into an "egg-chicken-egg" problem?
> > - About the ablation studies with $\lambda$ is limited, how do you set the range of the hyperparameter? Why are there only two hyperparameter choices for GOOD-CMNIST-color?

---

> > > ### Author Response · Authors · 2024-11-26
> > > **Response to Additional Comment by Reviewer FKFr**
> > >
> > > > Could you elaborate more on why the classifier could capture the semantic information?
> > >
> > > We appreciaite the question to help clarify how our conditional reverse-time SDE can maintain prior label knowledge.
> > >
> > > **Prior Knowledge of Out-of-Distribution (OOD) Samples**: As outlined in [1], our prior knowledge about the out-of-distribution aspect of input graphs $G$ is that they comprise two distinct components: the causal variable $C$ and the non-causal variable $S$, such as the House motif and the Tree base in the GOOD-Motif dataset. Here, $C$ serves as the sole endogenous parent crucial for determining the ground-truth label $y$, while $S$ is an element that varies across different environments. When synthesizing OOD samples, we use the label variable $y$ as prior knowledge and conditions to control the diffusion procudure, detailed as follows.
> > >
> > > **Conditional Reverse-time SDE with Label Knowledge**: Our approach specifically maintains the integrity of the causal part $C$ while exploring the unstable non-causal part $S$ through the application of a conditional reverse-time Stochastic Differential Equation (SDE), as presented in Eq. (6). In Eq. (6), the condition $y_G$ controls the generated graph to maintain the label knowledge (i.e., keeping causal part $C$), and the condition $y_\text{ood}$ augments the graph to explore the environment (i.e., varying non-causal part $S$). The derivation in our paper illustrates that the gradient of the log probability for this conditional reverse diffusion, as expressed in Eq. (9). **We use the classifier to model $p_t\left(\mathbf{y}_G \mid G_t\right)$ in Eq. (9).**
> > >
> > > Eq. (9) is featured with the following properties:
> > >
> > > **Preserving the Label Knowledge by $y_G$**: The second term $\log p_t\left(\mathbf{y}_G \mid G_t\right)$ aims to maximize the prediction probability of the graph's ground-truth label $y_G$ for each step. This essentially maintians the prior label knowledge when synthesizing the OOD samples.
> > >
> > > **Exploring the Training Distribution by $y_\text{ood}$**: Combining two terms, the conditional score function creates a marginal distribution proportional to $p_t\left(G_t, \mathbf{y}_G\right)^{1-\sqrt{\lambda}}$.
> > >
> > > When environmental graphs are explored, their patterns exist in the low-density regions of the original distribution. Conversely, un-explored graphs exist in high-density regions. With exploration strength $\lambda=0$, the marginal distribution becomes $p_t\left(G_t, \mathbf{y}_G\right)$,
> > >
> > > which denoises the synthesized graphs from the augmented distribution to follow the original distribution $P_{\operatorname{tr}}(G, Y)$, thus both casual part $C$ and non-casaul part are preserved. By adjusting $\lambda$, we regulate dispersion, allowing generated graphs to maintain the causal $C$ and explore non-causal $S$, thus increasing the divergence between $P_{\mathrm{tr}}\left(G, Y \mid \mathbf{y}_{\text {ood }}=\lambda\right)$ and
> > >
> > > $P_{\mathrm{tr}}(G, Y)$.
> > >
> > >
> > >
> > > > As the classifier is used for both identifying the semantic information and instructing the diffusion process, would not the whole training turn into an "egg-chicken-egg" problem?
> > >
> > > Our classifier is used solely for preserving label knowledge in the synthesized OOD samples and not for identifying semantic information. Additionally, we have provided results demonstrating that the classifier alone cannot effectively address covariate shifts.
> > >
> > >
> > > > About the ablation studies with $\lambda$ is limited, how do you set the range of the hyperparameter?
> > >
> > > As demonstrated in [2], exploration must remain within a meaningful deviation range from the original distribution to avoid disrupting the causal component $C$.
> > > Hence, we started with a relatively small $\lambda$ to ensure that the exploration does not destroy the causal component. In addition, $\lambda$ must be carefully selected based on performance metrics from OOD validation sets across various datasets.
> > >
> > > > Why are there only two hyperparameter choices for GOOD-CMNIST-color?
> > >
> > > Because the causal components in GOOD-CMNIST-color are the entire graph structures, and the covariate shift occurs only in node features (node colors), we selected a relatively large $\lambda$ with a larger step size to adjust $\lambda$.
> > >
> > >
> > > **References**:
> > >
> > > [1] Wu, et al. Discovering invariant rationales for graph neural networks. https://arxiv.org/abs/2201.12872
> > >
> > > [2] Sui, et al. Unleashing the Power of Graph Data Augmentation on Covariate Distribution Shift. https://arxiv.org/abs/2211.02843

---

### Official Review · Reviewer_iEJh · 2024-11-04

**Soundness:** 2
**Presentation:** 3
**Contribution:** 3
**Rating:** 3
**Confidence:** 3

**Summary:**

The paper proposes OODA, a score-based OOD augmentation framework for graph neural networks aimed at enhancing generalization under covariate shift conditions. It leverages a score-based generative model using diffusion processes to synthesize new graph environments, retaining stable predictive patterns while introducing controlled shifts. It generates OOD graphs that maintain graph validity and stable patterns without requiring explicit environmental separation. The authors conduct experiments across synthetic and real-world datasets, demonstrating OODA's advantages over traditional augmentation and invariant learning methods.

**Strengths:**

1. The research topic graph OOD is important. The idea of using OOD graphs with diffusion for simulating the environment is interesting.
2. The experiments with synthetic and real-world datasets show effectiveness, which is good.
3. The method is easy-to-understand overall.

**Weaknesses:**

1. One of the main concern is the motivation of the method, which I think is not well explained. I do not understand why the diffusion processes can synthesize OOD samples without enough and well-defined prior knowledge of the out-of-distribution.
2. The methodology to adjust shifts via hyperparameters, such as λ. However, it lacks a thorough discussion on selecting these parameters across different datasets, impacting reproducibility and usability.
3. The discussions on graph OOD are very limited, especially the Invariant Graph Learning part. The authors might consider referring to the survey [1] or the other surveys for introducing more related works and comparing the differences.

[1] Li, et al. Out-Of-Distribution Generalization on Graphs: A Survey. https://arxiv.org/pdf/2202.07987

I am happy to increase my score if the concerns can be fully addressed.

**Questions:**

See Weaknesses.

---

> ### Author Response · Authors · 2024-11-23
> **Reponse to  Reviewer iEJh -- Part 1**
>
> Thank you for your insightful comments and requests for clarifications. We will try addressing your concern below:
>
> > **W1**: Question on motivation: why the diffusion processes can synthesize OOD samples without enough prior knowledge of the OOD.
>
>
> We appreciaite the question to help clarify how our conditional reverse-time SDE can maintain prior label knowledge and explore OOD environment simultaneously.
>
> **Prior Knowledge of Out-of-Distribution (OOD) Samples**: As outlined in [1], our prior knowledge about the out-of-distribution aspect of input graphs $G$ is that they comprise two distinct components: the causal variable $C$ and the non-causal variable $S$, such as the House motif and the Tree base in the GOOD-Motif dataset. Here, $C$ serves as the sole endogenous parent crucial for determining the ground-truth label $y$, while $S$ is an element that varies across different environments. When synthesizing OOD samples, we use the label variable $y$ as prior knowledge and conditions to control the diffusion procudure, detailed as follows.
>
> **Conditional Reverse-time SDE with Label Knowledge**: Our approach specifically maintains the integrity of the causal part $C$ while exploring the unstable non-causal part $S$ through the application of a conditional reverse-time Stochastic Differential Equation (SDE), as presented in Eq. (6). The condition $y_G$ controls the generated graph to maintain the label knowledge (i.e., keeping causal part $C$), and the condition $y_\text{ood}$ augments the graph to explore the environment (i.e., varying non-causal part $S$). The derivation in our paper illustrates that the gradient of the log probability for this conditional reverse diffusion is expressed in Eq. (9).
>
> which is featured with the following properties:
>
> **Preserving the Label Knowledge by $y_G$**: The second term $\log p_t\left(\mathbf{y}_G \mid G_t\right)$ aims to maximize the prediction probability of the graph's ground-truth label $y_G$ for each step. This essentially maintians the prior label knowledge when synthesizing the OOD samples.
>
> **Exploring the Training Distribution by $y_\text{ood}$**: Combining two terms, the conditional score function creates a marginal distribution proportional to $p_t\left(G_t, \mathbf{y}_G\right)^{1-\sqrt{\lambda}}$.
>
> When environmental graphs are explored, their patterns exist in the low-density regions of the original distribution. Conversely, un-explored graphs exist in high-density regions. With exploration strength $\lambda=0$, the marginal distribution becomes $p_t\left(G_t, \mathbf{y}_G\right)$,
>
> which denoises the synthesized graphs from the augmented distribution to follow the original distribution $P_{\operatorname{tr}}(G, Y)$, thus both casual part $C$ and non-casaul part are preserved. By adjusting $\lambda$, we regulate dispersion, allowing generated graphs to maintain the causal $C$ and explore non-causal $S$, thus increasing the divergence between $P_{\mathrm{tr}}\left(G, Y \mid \mathbf{y}_{\text {ood }}=\lambda\right)$
>
> and $P_{\mathrm{tr}}(G, Y)$.
>
> **Balancing Label Preservation and Environment Exploration**: As demonstrated in [2], exploration must remain within a meaningful deviation range from the original distribution to avoid disrupting the causal component $C$. Hence, $\lambda$ must be carefully selected based on performance metrics from OOD validation sets across various datasets.
>
> > **W2**: Discussion on selecting hyperparameter $\lambda$ across different datasets, impacting reproducibility and usability.
>
> **A2**: We determine the value of $\lambda$ by evaluating its performance on OOD validation sets across different datasets, as detailed in Appendix A.4.2. This section provides a comprehensive discussion on our selection criteria, which ensures that the chosen $\lambda$ optimizes model performance while maintaining generalizability. Additionally, we explore the impact of varying $\lambda$ on navigating the space of the original distribution, as illustrated in Figures 2, 4, and 5. These figures demonstrate how different $\lambda$ values influence both the preservation and exploration of distributional shifts, thereby providing insights into its role in our methodology. This thorough analysis enhances the reproducibility and applicability of our approach across diverse datasets.

---

> > ### Author Response · Authors · 2024-11-23
> > **Reponse to Reviewer iEJh -- Part 2**
> >
> > > **W3**: Referring surveys for more discussion and comparisons on related works in graph OOD.
> >
> > Thank you for the suggestion. Graph OOD is indeed an extensively studied area, and we have extended our discussion on related works in the modified version. We previously employed a similar set of state-of-the-art baseline methods, backbones, and hyperparameter-tuning techniques, as described in recent works [2] and [3]. In addition to these, we now extend our related work section to include a discussion of the survey [5] and latest advancements referenced in [3] and [4]. This comprehensive overview ensures our methodology aligns with leading standards while also integrating recent developments in the field.
> >
> > **References:**
> >
> > [1] Wu, et al. Discovering invariant rationales for graph neural networks. https://arxiv.org/abs/2201.12872
> >
> > [2] Sui, et al. Unleashing the Power of Graph Data Augmentation on Covariate Distribution Shift. https://arxiv.org/abs/2211.02843
> >
> > [3] Li, et al. Graph structure extrapolation for out-of-
> > distribution generalization. https://arxiv.org/abs/2306.08076
> >
> > [4] Huang, et al. Enhancing size generalization in graph
> > neural networks through disentangled representation learning. https://arxiv.org/abs/2406.
> >
> > [5] Li, et al. Out-Of-Distribution Generalization on Graphs: A Survey. https://arxiv.org/pdf/2202.07987

---

> > > ### Comment · Reviewer_iEJh · 2024-11-29
> > >
> > > Thank you to the authors for providing a detailed response.
> > >
> > > - The reply to **W3** has addressed my concern. I agree with the authors that "Graph OOD is indeed an extensively studied area." This is precisely why I suggested expanding the discussion in the related works section to provide more context.
> > >
> > > - The reply to **W2** does not fully resolve my concerns, particularly regarding the current experiments on the impact of $\lambda$.
> > >
> > > - The reply to **W1** remains unaddressed. Could you provide an example using GOOD-CMNIST to demonstrate how to synthesize the OOD samples (e.g., the sixth color) from the samples with the first five colors? I believe this would help clarify the motivation behind the approach.

---

> > > > ### Author Response · Authors · 2024-11-29
> > > > **Response to Additional Comment by Reviewer iEJh**
> > > >
> > > > > The reply to W2 does not fully resolve my concerns, particularly regarding the current experiments on the impact of $\lambda$.
> > > >
> > > > The current experiments on the impact of $\lambda$ aim to verify its effect on controlling the distance between the augmented distribution and the original distribution. Additionally, we provide examples of the generated out-of-distribution (OOD) graphs in Table 3, offering further qualitative evidence of the impact of $\lambda$.
> > > >
> > > > In addition to the current experiments, we conducted new experiments to assess the impact of $\lambda$ on model performance.
> > > >
> > > >
> > > > **Sensitivity of hyperparameter $\lambda$**: As detailed in Appendix A.4.2, we determine the optimal value of $\lambda$ by evaluating its effectiveness on out-of-distribution (OOD) validation sets across various datasets. Below, we provide ablation results that further illustrate the sensitivity of our method to different $\lambda$ values:
> > > >
> > > >
> > > > | $\lambda$ | GOOD-Motif-base | GOOD-HIV-scaffold |
> > > > | -------- | -------- | -------- |
> > > > |  0.01   |   91.80   | 78.45   |
> > > > |  0.02   | 92.97     | 78.50     |
> > > > |  0.03   | 92.83     | 79.49     |
> > > > |  0.04   | 93.03     | 78.71     |
> > > > |  0.05   | 92.77    | 78.69   |
> > > >
> > > >
> > > > | $\lambda$ | GOOD-CMNIST-color |
> > > > | -------- | -------- |
> > > > |  0.05   |   68.66   |
> > > > |  0.1   | 67.91     |
> > > >
> > > > We hope these experiments provide clarity on the selection of $\lambda$ across different datasets.
> > > >
> > > > > Could you provide an example using GOOD-CMNIST to demonstrate how to synthesize the OOD samples (e.g., the sixth color) from the samples with the first five colors?
> > > >
> > > > In one graph sample from GOOD-CMNIST, the causal components are represented by the entire graph structure, while the covariate shift occurs only in the node features (specifically, the node colors). For instance, let's denote $\mathbf{y}_G = 0$ to indicate that the digit represented in the graph sample is '0', with the graph structure embodying this digit.
> > > >
> > > > When $\lambda = 0$, the conditional score function in Eq. (6) results in a marginal distribution $p_t\left(G_t, \mathbf{y}_G=0\right)$
> > > >
> > > > that closely resembles the original data distribution $P_{\operatorname{tr}}(G, Y=0)$. Consequently, the reverse-time diffusion process denoises the perturbed graphs to a distribution conditioned solely on $\mathbf{y}_G = 0$. As a result, the graph samples retain the structure of the digit '0' while including the first five colors.
> > > >
> > > > In the original distribution, data for the class $Y=0$ typically forms a tightly centered cluster within the data space, characterized by well-defined causal relationships. When we increase $\lambda$, it modulates this distribution to enhance variability and exploration.
> > > >
> > > > As $\lambda$ increases, the conditional score function in Eq. (6) produces a broader distribution compared to the original data distribution $P_{\operatorname{tr}}(G, Y=0)$. This increased dispersion allows for the sampling of graphs that maintain the digit '0' structure while also incorporating more diverse and novel non-causal node colors. The sixth color is highly likely to emerge from this broader distribution, allowing for a wider array of colors and fostering exploration beyond the tightly clustered native structures. This process can lead to novel insights and enhance robust model learning.

---

### Meta-Review · Area_Chair_VaZ5 · 2024-12-20

**Metareview:**

(a) The paper introduces OODA, a framework designed to enhance the out-of-distribution (OOD) generalization capabilities of graph neural networks. Using score-based graph generation strategies, the method was tested across various datasets, demonstrating improved effectiveness over traditional approaches.

(b) The reviewers find the application of diffusion processes for graph OOD generalization novel and appreciate the validation of the framework's effectiveness through experiments across various scenarios.

(c) The paper does not clearly explain why diffusion processes are suitable for generating OOD samples without extensive prior knowledge of the OOD space. Several reviewers found the selection of hyperparameters like $\lambda$, crucial for adjusting shifts, confusing. Additionally, the necessity of training a classifier to guide OOD generation raised concerns about the validity of the generated samples, highlighting a potential chicken-and-egg problem depending on the classifier's effectiveness.

(d) The reviewers express reservations about the paper, citing a lack of theoretical support, increased computational demands, and potential unfair comparisons due to the use of graph transformers instead of traditional GNN architectures for the proposed method.

**Additional Comments On Reviewer Discussion:**

Reviewers expressed ongoing concerns about several critical issues:

- The unclear motivation for using diffusion processes and inadequate discussion on hyperparameter selection.
- Limited novelty and unclear preservation of features.
- Concerns regarding graph diversity and the method's effectiveness in size shifts.
- A lack of theoretical support.

While the authors have attempted to address these issues, it remains uncertain whether all these points can be thoroughly clarified without substantial revisions.

Based on the reviewers' comments and assessments, which I find appropriate, I have decided to follow the majority opinion.

---

### Decision · Program_Chairs · 2025-01-22

Reject